https://doi.org/10.1038/s42003-022-03969-7　**OPEN**
# Contribution of CRISPRable DNA to human complex traits

Ranran Zhai [1,2], Chenqing Zheng [1], Zhijian Yang [1,2], Ting Li [1,2], Jiantao Chen [1,2] & Xia Shen [1,2,3,4]✉

CRISPR-Cas is a powerful genome editing tool for various species and human cell lines, widely used in many research areas including studying the mechanisms, targets, and gene therapies of human diseases. Recent developments have even allowed high-throughput genetic screening using the CRISPR system. However, due to the practical and ethical limitations in human gene editing research, little is known about whether CRISPR-editable DNA segments could influence human complex traits or diseases. Here, we investigated the human genomic regions condensed with different CRISPR Cas enzymes' protospacer-adjacent motifs (PAMs). We found that Cas enzymes with GC-rich PAMs could interfere more with the genomic regions that harbor enriched heritability for human complex traits and diseases. The results linked GC content across the genome to the functional genomic elements in the heritability enrichment of human complex traits. We provide a genetic overview of the effects of high-throughput genome editing on human complex traits.

[1] Biostatistics Group, School of Life Sciences, Sun Yat-sen University, Guangzhou, China. [2] Center for Intelligent Medicine Research, Greater Bay Area Institute of Precision Medicine (Guangzhou), Fudan University, Guangzhou, China. [3] State Key Laboratory of Genetic Engineering, School of Life Sciences, Fudan University, Shanghai, China. [4] Centre for Global Health Research, Usher Institute, University of Edinburgh, Edinburgh, UK. ✉email: shenx@fudan.edu.cn

The CRISPR–Cas system is originally a prokaryotic immune system that can provide acquired resistance to foreign genetic elements from phages[1]. Since its discovery, CRISPR–Cas has been modified to be a powerful genome editing tool for various species and cells, including human cells[2,3]. Due to their high precision and low cost, CRISPR systems have been widely used in many research areas, including discovering mechanisms of human diseases, identifying disease targets, and gene therapy[4–6]. Besides, the CRISPR–Cas systems also have been used in high-throughput genetic screening for genes[7] and functional regulatory elements[8–11] involved in various biological processes. Recent development has extended the use of the CRISPR–Cas system for so-called "prime editing" (PE), which can write new genetic information to a specific target site[12]. Saturation prime editing (SPE)[13] has succeeded in functionally assaying hundreds of single nucleotide variants in both the *NPC1* and *BRCA2* gene, further advancing the field toward precision medicine.

There are two different classes of CRISPR–Cas system: class 1 (including type I, III, and IV), which uses multiple Cas enzymes as effector nucleases, and class 2 (including type II, V, and VI), which uses a single Cas enzyme[14], so that class 2 CRISPR–Cas systems are easier to manipulate. Among the class 2 systems, the type II CRISPR-Cas9 and type V CRISPR-Cas12a (previously known as CRISPR-Cpf1) are most characterized. A classic CRISPR-Cas9/Cas12a system contains a single guide RNA (sgRNA) that defines the target sequence and a Cas as an effector nuclease to induce a double-strand break (DSB), resulting in insertion or deletion at the target site. As effectors of the CRISPR systems, Cas9 introduces blunt DSBs after cleavage[2], while Cas12a introduces DSBs with a short overhang[15].

Another important difference between the two types of Cas is that they require distinct protospacer-adjacent motifs (PAMs), which are essential for binding and cleaving the up- or downstream target sequences. For example, Cas9 from *Streptococcus pyogenes* (SpCas9) needs a 5′-NGG-3′ (N stands for A, T, G, or C) PAM downstream of the target region[2,16], while Cas12a from *Acidaminococcus* sp. (AsCas12a) requires a 5′-TTTV-3′ (V stands for A, G, or C) PAM upstream[17,18]. Such requirements have limited the utility and flexibility of the CRISPR–Cas systems.

Efforts have been made to expand the PAM recognition space by discovering natural Cas orthologs and modifying well-characterized Cas proteins. Besides SpCas9, a smaller Cas9, SaCas9[19] from *Staphylococcus aureus* and its variant KKH-SaCas9[20] have been identified and showed robust genome editing activities at endogenous human target sites. Other Cas9 orthologs with varied protein size and PAM specificity have emerged in addition to SpCas9 and SaCas9[21–24]. Meanwhile, with the better characterization of SpCas9, mutated SpCas9 variants[25–28] have provided an even wider target range, especially the near-PAM-less SpRY variant developed by Walton et al.[29] that can theoretically recognize every single base. Also, several Cas12a variants, including the enAsCas12a[30], RVR- and RR-AsCas12a[31], and FnCas12a[32] from the *Francisella novicida* have also been modified to use less T-rich PAMs. These enzymes altogether have enabled researchers to edit a large proportion of the human genome via CRISPR. However, little is known about whether editing such "CRISPRable" genomic regions may potentially influence human complex traits or diseases.

Massive genome-wide association studies (GWAS) have helped better understand the genetic architecture of human complex traits and diseases. GWAS can identify genetic variants (usually single nucleotide polymorphisms, SNPs) that are associated with particular phenotypes. In recent years, human genetics analysis has emphasized the use of GWAS summary statistics to provide more insights into complex trait biology[33]. For example, GWAS

summary statistics can be used for estimating SNP heritability and genetic correlation across human phenotypes[34–36], as well as the enrichment of complex trait heritability at particular annotated regions in the genome[34,37].

Here, utilizing the GWAS summary statistics resource, we evaluate the genetic contribution of the CRISPRable genomic regions, featured with 21 different Cas enzymes, to 28 human complex traits and diseases. By partitioning the heritability of these phenotypes, we found that Cas enzymes using GC-rich PAMs have more potential to influence human complex traits. These CRISPRable pieces of the genome have higher odds of overlapping with coding regions, enhancers, promoters, and other functional regulatory elements. Our results provide a landscape of Cas-featured CRISPRable DNA segments contributing to human complex traits, providing insights into future high-throughput CRISPR screening studies.

## Results

### Heritability of human complex traits is enriched in Cas/PAM-enriched genomic regions.
We compiled a list of 21 Cas enzymes, consisting of 16 Cas9 and 5 Cas12a, covering 77 unique PAM sequences (Supplementary Data 1). These Cas enzymes include both natural Cas enzymes and their engineered variants with altered PAM specificity. In addition, their PAMs vary in length and base composition, making them a good representative subgroup of the CRISPR–Cas systems used in genome editing.

To determine which genomic regions are most editable, we first divided the human autosomes into 20,000 segments, each with a length of about 144 kilobases (kb). We counted the number of each specific PAM sequence within each segment, then denoted the top 2000 segments (10% of the genome) that have the highest number of specific PAM sequences as PAM-enriched regions. In the context that one Cas could recognize more than one PAM sequence, considering all its potential PAMs is more reasonable when choosing the proper CRISPR–Cas platform. Thus, in the subsequent analysis, we determined the most editable regions of each Cas by summing the number of all its potential PAMs within each genome segment (removing duplications between PAMs) and denoting the top 2000 segments (10% of the genome) as Cas-enriched regions.

To investigate the magnitude of these Cas-enriched regions' contribution to human complex traits, we applied stratified linkage disequilibrium (LD) score regression (S-LDSC)[34,37] to partition the heritability of each human complex trait. Heritability enrichment was defined as the proportion of heritability explained by the annotated SNPs divided by the proportion of SNPs annotated. For each Cas, we annotated common SNPs within the top 2000 enriched regions, resulting in 21 sets of binary annotations, where each annotation contains about 11% of the total 9,997,231 common SNPs across the autosomal genome. Here on, for simplicity, we name each annotation by its corresponding Cas enzyme (e.g., SpCas9 stands for the SNPs annotated within the SpCas9-enriched regions).

We examined 28 traits, including five anthropometric traits, seven common diseases, six mental disorders, six metabolites measurements, and four other lifestyle phenotypes (Fig. 1 and Supplementary Data 2 and 3). Across all annotations, we found that annotations for Cas enzymes (Cas9 or Cas12a) that have higher GC content in their PAMs have higher heritability enrichment for most complex traits. Furthermore, (1) levels of heritability enrichment vary with the categories of traits, indicating different genetic regulatory mechanism of different types of phenotypes; (2) three Cas annotations with high-GC content show the highest enrichment in three immunological diseases, including ulcerative colitis (max enrichment = 3.5, $P = 8.6 \times 10^{-9}$ on the St3Cas9 annotation), inflammatory bowel

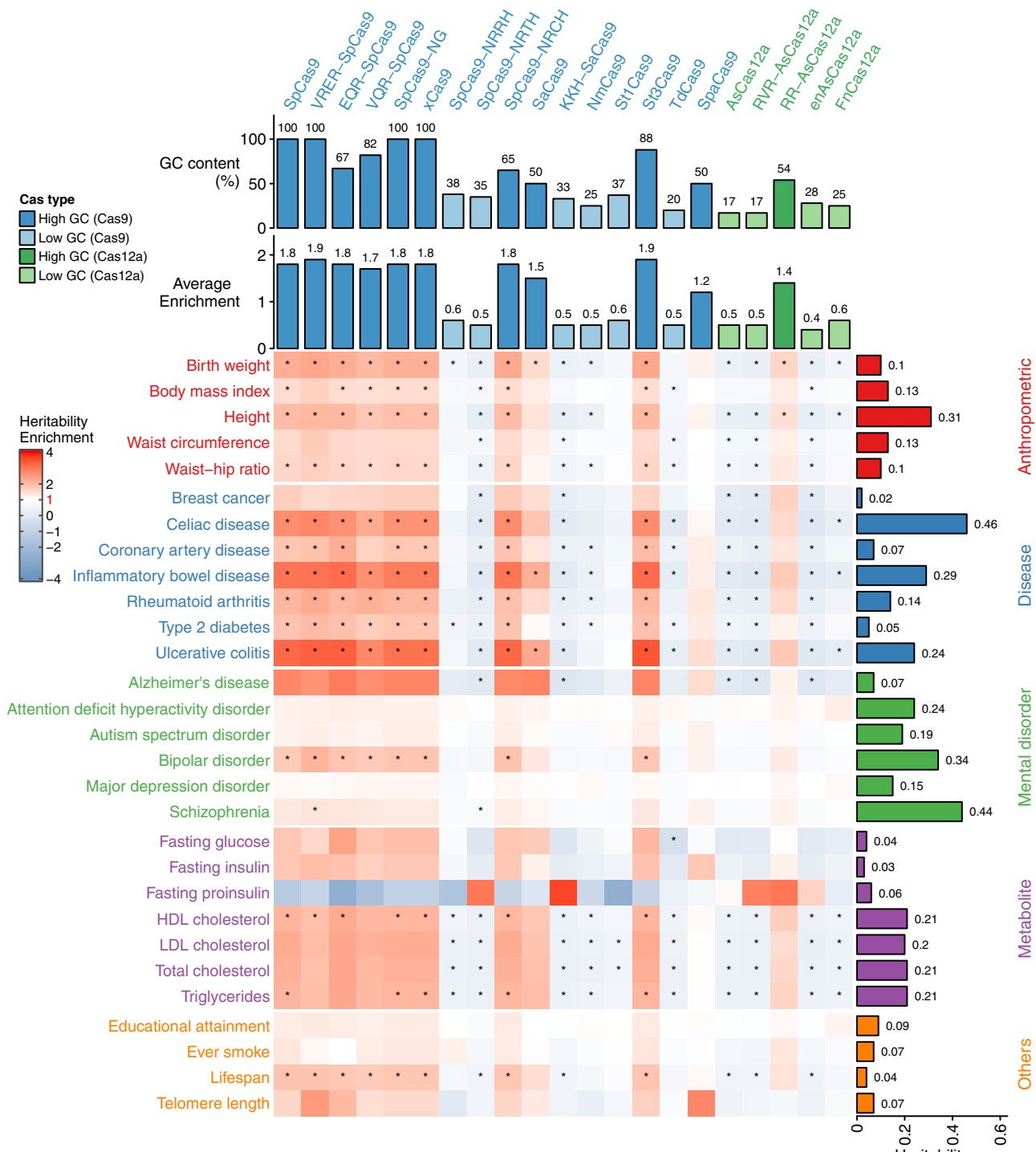

**Fig. 1 Heritability enrichment of 28 human complex traits in the Cas-enriched genomic regions.** The heatmap shows the heritability enrichment of Cas annotations on 28 traits. Traits are categorized into five domains and their names are in different colors. The barplot right to the heatmap shows the SNP base heritability estimated by LDSC. The two bar plots on top of the heatmap show the overall GC content (top) and average heritability enrichment (bottom) for $n = 21$ Cas enzymes, respectively. *Indicates significance after multiple testing correction with a false discovery rate of 0.05.

disease (max enrichment = 3.2, $P = 2.5 \times 10^{-9}$ on the St3Cas9 annotation), and celiac disease (max enrichment = 2.8, $P = 2.7 \times 10^{-7}$ on the VRER-SpCas9 annotation); (3) mental disorders have less enriched heritability in the Cas/PAM annotations than the other traits, except for Alzheimer's disease (max enrichment = 3.0, $P = 0.07$ on the EQR-SpCas9 annotation) and bipolar disorder (max enrichment = 2.2, $P = 2.2 \times 10^{-11}$ on the VRER-SpCas9 annotation).

**Cas enzymes with higher PAM GC content tend to annotate more heritability of human complex traits.** When classifying the 21 Cas enzymes into four categories by their overall PAM GC content and Cas types, we found a clear trend that the higher the GC content of the Cas enzyme PAM(s) is, the more heritability was captured by the corresponding annotated regions (Fig. 1). When investigating the annotated regions of different categories of Cas enzymes, we found that enriched regions of Cas with a

higher overall GC content were more condensed on chromosomes 16, 17, 19, and 22, even after adjusting for chromosome length (Supplementary Fig. 1a, b). The same chromosomes also have higher GC content and gene density compared to other autosomes (Supplementary Fig. 1c), which was previously elaborated[38,39], indicating that genomic regions with high-GC content might be one source that led to the enriched heritability of human complex traits that we observed.

We next compared the GC count and PAM count within each segment for every individual PAM. We found that for PAMs that have a high-GC content, their top 2000 PAM-enriched segments are more likely to be in the top 2000 GC-rich segments (Pearson's $r = 0.86$, $P = 1.5 \times 10^{-23}$) (Supplementary Figs. 2–77). But there are exceptions; for instance, the PAM GCAA, with a GC content of 50%, has 1993 segments (out of 2000) that did not overlap with the top GC-rich segments. Meanwhile, among the 2000 top segments of GGCG annotation, 666 did not belong to the top GC-rich segments, despite that GGCG having a GC content of 100% (Fig. 2).

**Annotations of Cas enzymes with higher PAM GC content capture-enriched functional genomic elements.** To further investigate the Cas annotations, we also compared them with other established functional annotations. The functional annotations include gene coding regions, epigenetic markers (e.g., methylation and acetylation of the histone), and other regulatory elements such as promoters and enhancers. We found that SNPs annotated for high-GC content Cas enzymes have higher odds of being also annotated for coding regions, gene expression regulatory elements such as acetylation of histone H3 at lysine 9 (H3K9ac) and lysine 27 (H3K27ac), transcription start sites (TSS), and enhancers/promoters identified by different sources (Fig. 3). These functional annotations were also previously shown to harbor enriched heritabilities of nine human complex traits[37]. By contrast, other Cas annotations have much lower odds of being annotated within these functional regions.

Besides the autosomes, the X chromosome is of importance and complexity in the genetic regulation of human complex traits and diseases. For instance, mutations in the histone demethylase gene *KDM6A* could cause the Kabuki syndrome[40]; while the SNPs within the *HS6ST2* gene are associated with neuroticism[41]. At present, the S-LDSC tool we used to assess heritability enrichment did not include the X chromosome, as no X chromosome functional annotations were available for the analysis. To investigate whether the pattern we found in the autosomal regions was similar on the X chromosome, we used the SNP annotation tool, SNPnexus[42], to annotate the 125,497 HapMap3 SNPs on the X chromosome. We included four gene annotations from UCSC[43], eight epigenetic markers from Roadmap[44], and five regulator elements from Ensembl[45] in the analysis. Consistent with the autosomal genomic region, we found that SNPs annotated for high-GC content Cas also had higher odds of being annotated for coding regions, H3K27ac, promoter, and so on (Fig. 4). This consistency indicates that regions enriched for Cas with high-GC content on the X chromosome are also likely to harbor enriched heritability for human complex traits.

## Discussion

We examined the heritability enrichment levels of the PAM-enriched genomic regions of 21 Cas on 28 human complex traits. We identified heritability enrichment for regions with high-GC Cas PAMs across these traits, especially the immunological diseases. Regions with enriched heritability also showed substantial overlap with functional annotations such as gene coding regions, TSS, and other regulatory elements.

Although most of the genome has A/T rather than G/C bases, there are genome pieces that harbor condensed GC content. Our results showed that high-GC content Cas enzymes have a higher potential for interfering with human complex traits and diseases. This potential relies on the fact that high-GC Cas enzymes have greater access to GC-rich regions, which are in turn enriched for genes[39], TSS[46], promoters[47], and enhancers[48] that contribute to human complex traits. Beyond that, we provide evidence that GC-rich regions are enriched for other functional regulatory elements such as histone methylation, linking them with elevated heritability for human complex traits and diseases.

Heritability enrichment for high-GC PAMs, such as the initially characterized SpCas9 NGG PAM, therefore closely aligns with known feature enrichment for high-GC genome sequences[39,46–48], although such prior knowledge did not enter our statistical analyses, with its focus on the heritability of complex traits. Given the relationship between elevated GC content and coding sequences in established literature and the enrichment of complex trait heritability in coding regions, one might infer that GC content itself was driving the observed heritability enrichment. This might make the enrichment of complex trait heritability in GC-enriched regions seem obvious, but no literature systematically analyzed this using the post-GWAS data. Our results thus also bring together genomic regions with high-GC content and the established functional annotations in terms of heritability enrichment of human complex traits. Our analysis seems to reveal that heritability enrichment on different functional genomic annotations could mostly be driven or explained by GC content. The results are, besides their relevance to CRISPR, also useful in general for other complex trait genetics studies.

Furthermore, each short PAM can exist nearly everywhere along the genome, and a specific PAM does not only target a GC-rich piece in the genome. We tried to demonstrate the distribution of GC content versus the distributions of different types of PAMs. Although GC content itself substantially explains the heritability enrichment of complex traits, there can still be a piece of the genome that harbors a high number of a GC-rich PAM sequence which is not one of the top GC-rich regions. In the practice of CRISPR–Cas techniques, we suggest that one should choose carefully to balance the PAM specificity and GC content, for both high editing coverage and noticeable changes in genetic effects.

The broadened PAM recognition space has made the CRISPR–Cas systems powerful in manipulating and investigating the human genome. However, the PAM composition is not the only factor to be considered in the selection of Cas enzymes. One of the most important aspects is the trade-off between wider target space and increased off-target activity, which is usually introduced by the mismatch tolerance of Cas enzymes in the existence of similar sequences to the actual target. Especially in the existence of the near PAM-less Cas9 variant, cautions are needed to balance the wide target range and potential off-target effects. With this regard, we matched 8,956 off-target sites from 13 studies to our defined 20,000 segments. We found that the more NGG that a segment has, the more off-target sites are within that segment, which fits the theory that PAM-riched regions have an increased propensity for off-targeting (Supplementary Fig. 78). However, the cleavage frequency is not correlated with the number of NGG of the corresponding segments, suggesting that the cleavage process itself is not solely dependent on the presence of PAM. Although the lack of off-target effect evaluation sources of different PAMs across the human genome made it difficult to consider the off-target effects in the current analysis, the

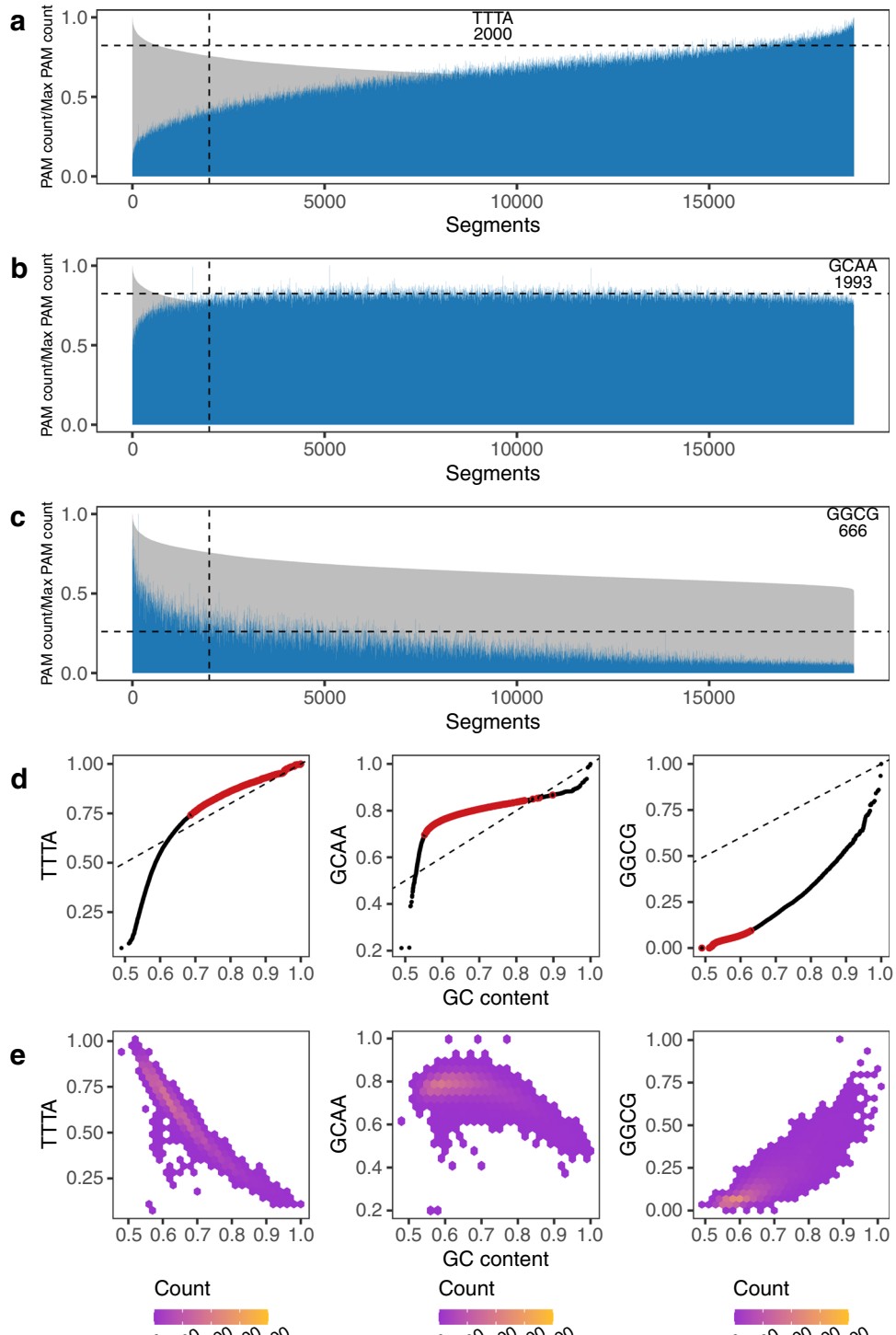

**Fig. 2 TTTA, GCAA, and GGCG enriched regions compared with GC-rich regions. a–c** TTTA, GCAA, and GGCG content of 18,763 segments (excluding 1237 segments with all-N bases), respectively. The genome segments are ordered by decreasing GC content (gray), and the corresponding PAM content is plotted in the same order (blue). The segments left to the vertical dashed line are the top 2000 segments for GC content, and the segments above the horizontal dashed line are the top 2000 segments for the TTTA, GCAA, and GGCG PAM, respectively. The number in the plot shows the number out of the top 2000 PAM segments that are not in the top 2000 GC-rich segments. **d** Quantile–quantile plot of the segments GC content v.s. PAM content distributions, where the red circles show the top 2000 PAM segments, and the dashed line represents $y = x$. **e** Hexbin plot of segments GC v.s. PAM content, where the color represents the count of points in each hexagon. GC and PAM content is normalized to the 0–1 range.

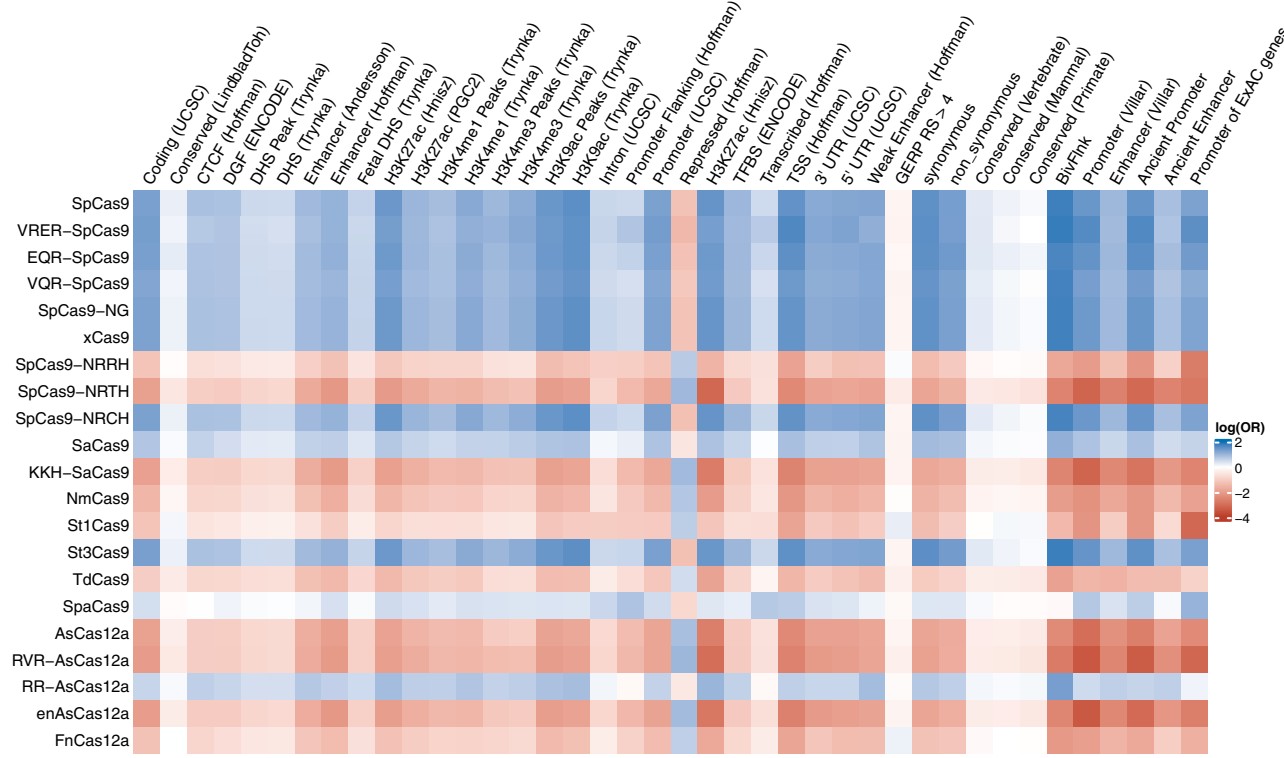

**Fig. 3 Cas using GC-rich PAMs shows more overlap with functional annotations.** The rows represent Cas annotations, while the columns represent functional annotations (see Supplementary Data 4 for details). The odds ratio (OR) between pairs of Cas annotation and functional annotation was calculated by Fisher's Exact test on $n = 9,997,231$ SNPs. The plot shows log(OR), in which blue indicates OR >1.

utilization of high-fidelity Cas enzymes could potentially reduce such tolerance and off-target effects[29,30].

With the recent development of prime editing (PE)[12] and saturation prime editing (SPE)[13], our results provide a general landscape for high-throughput genome editing that, when a particular PAM is widely applied across the human genome, more genetic effects could be altered or introduced if the PAM contains more G/C bases. Although systematic validations of such gene editing effects on complex traits in a human population are practically and ethically impossible, the results can provide useful insights in specific research using human cell lines and even in other species when studying complex phenotypes.

## Methods

**Segmenting the human autosomes.** Analysis of the human genome was based on the GRCh37 assembly. To simplify the process, we considered 22 autosomal chromosomes end to end as one single piece with a total of 2,881,033,286 bases. We cut the entire piece into 20,000 segments, resulting in about 144,052 bases per segment. Using the fasta file, we get the base composition of each segment, as well as the GC content. We counted the number of PAM sequences in each segment, where the reverse strand was also considered. For example, when counting the number of the NGG sequence, we counted the number of both 5'-NGG-3' and 5'-CCN-3'. Only PAMs explicitly pointed out as efficient in the original reports were included in our analysis. Out of 20,000 segments, 1237 are all-N bases, which are excluded in the subsequent analysis. For the segments that are partially N, the GC content and PAM number are adjusted for its non-N length (144,052 − number of N bases).

**Annotation of Cas-enriched regions.** The overall GC content of one Cas is the proportion of GC bases in all bases of all its PAM sequences, excluding the N base at the ends. For example, the SpCas9 (PAM: NGG) has a GC content of 100%, while the EQR-SpCas9 (PAM: NGAG) has a GC content of 67%. Annotation of Cas-enriched regions is based on the number of individual PAMs within each segment. For Cas with more than one PAM sequence, we selected the top 2000 segments that have the highest sum of all its PAMs, denoting Cas-enriched regions. It is worth mentioning that though xCas9 has three PAM sequences (NG, GAA, and GAT)[27], we only count the number of NG PAM in

each segment as the sorting criteria when choosing the "Cas-enriched regions", for the count of GAA and GAT are already included when counting the NG PAM. The binary annotations of 21 Cas-enriched regions were made by the LDSC software. For each Cas annotation, out of 9,997,231 common SNPs used in the LDSC, SNPs within the selected top 2,000 genomic regions are annotated as 1, and other SNPs as 0.

**Heritability enrichment analysis.** LD scores for the annotated SNPs in each annotation were calculated based on a 1-centiMorgan (cM) window. Only Hap-Map3 SNPs were retained for the analysis; We used all 97 annotations from the "full baseline model" (v2.2) in the LDSC software as covariates in the stratified regression model. We used the –overlap-annot argument and frequency files 1000G_Phase3_frq.tgz to confine our analysis to SNPs with minor allele frequency (MAF) >5%. The MHC region was excluded from all analyses because of its unusual LD and genetic architecture.

**Summary association statistics of 28 traits.** The complex traits used for heritability enrichment analysis included 28 phenotypes, for which the large-scale genome-wide association meta-analysis had been conducted, with publicly available summary statistics. The 28 traits covered anthropometric measures, common diseases, mental disorders, plasma cholesterol levels, lifestyle, educational attainment, and so on. More details of the summary statistics sources are described in Supplementary Data 2.

**Annotation for the X chromosome.** For the X chromosome, we kept the segments in the same length (144 kb) as we did for the autosomes, resulting in 1070 segments in total. Top10% of the 1070 segments were selected as the Cas-enriched genomic regions on the X chromosome. In line with procedures followed in the analysis of autosomes, SNPs present in the top10% regions were annotated as 1, and other SNPs were annotated as 0 for each Cas. The functional annotations for the X chromosome include four gene annotations from UCSC, eight epigenetic markers from Roadmap, and five regulator elements from Ensembl, using the SNP annotation Tool SNPnexus. We queried 125,497 Hapmap3 SNPs on https://www.snp-nexus.org/v4/for these functional annotations.

**Statistics and reproducibility.** The significance of the heritability enrichment test was determined using the LDSC software, where a Wald test statistic was applied with jackknife-based standard errors. The significance of Pearson's correlation coefficient was determined based on Fisher's Z-transformed test statistic. The code

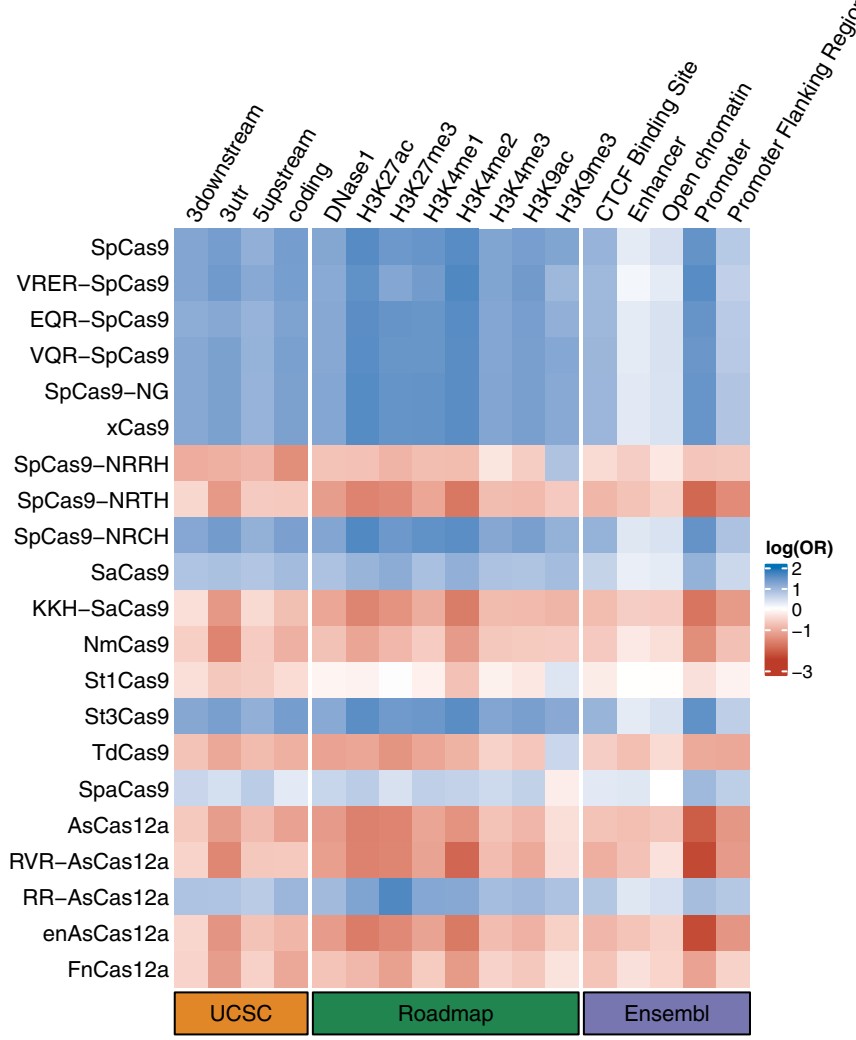

**Fig. 4 Cas using GC-rich PAMs also shows more overlap with functional annotations on the X chromosome.** The rows represent Cas annotations on the X chromosome, while the columns represent gene annotations (from UCSC), epigenomic markers (from Roadmap), and regulatory elements (from Ensembl). The odds ratio (OR) was calculated by Fisher's Exact test on $n = 125,497$ SNPs. The plot shows log(OR), in which blue indicates OR >1.

and source data to reproduce all the statistical analyses are provided (see "Data availability" and "Code availability").

**Reporting summary**. Further information on research design is available in the Nature Research Reporting Summary linked to this article.

## Data availability

GWAS summary statistics of 28 traits and diseases used in our studies are publicly available (see details in Supplementary Data 2). LDSC baseline model annotations v2.2 are available at https://storage.googleapis.com/broad-alkesgroup-public/LDSCORE/1000G_Phase3_baselineLD_v2.2_ldscores.tgz; crisprSQL database: http://www.crisprsql.com/index.php. Source data for graphs are available in figshare at https://doi.org/10.6084/m9.figshare.20455116.

## Code availability

LDSC software: https://github.com/bulik/ldsc; Code used for reproducing the analysis is available at Zenodo[49].

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

## Acknowledgements

X.S. received a grant (No. 12171495) from the National Natural Science Foundation of China. We thank Dr. Xiaoyan Jia (Fudan University) for his helpful comments.

## Author contributions

X.S. initiated and coordinated the study. R.Z. performed data analysis. C.Z. processed the sequencing data. Z.Y., T.L., and J.C. contributed to the analysis pipeline. R.Z. and X.S. wrote the paper. All authors approved the submitted version of the paper.

## Competing interests

The authors declare no competing interests.
