## [Peer Review File · Communications Biology]

Reviewers' comments:

Reviewer #1 (Remarks to the Author):

Summary:

Zhai et al. demonstrate an analysis of PAMs enrichment in human complex traits. The authors compare several different PAMs of various Cas enzymes and demonstrate that GC rich PAMs are more abundant in genomic regions that harbour enriched heritability for human complex traits.

Conclusions:

The relevance of this work is not clearly stated in the manuscript. The manuscript should be revised and some of the analyses might require further work (see comments).

Major concerns:

1. The general relevance of this study is unclear. It should be indicated right from the abstract – what is the relevance of the results. Currently, it seems that the PAM-annotations don't mean much more than a GC content analysis of human complex traits. The authors should elaborate whether there is an advantage to the enrichment they describe, where most, if not all genomic regions can be targeted by several gRNAs of any Cas variant (i.e. the probability of having a PAM sequence at the region of interest is very high, especially if using PAMless variants – see the next comment).

2. The limited PAM selection of 8 different PAMs is unclear. The authors should consider mentioning a recent work by Walton et al. ("Unconstrained genome targeting with near-PAMless engineered CRISPR-Cas9 variants") where the researchers developed Cas variants with a limited PAM requirement. Moreover, a list of 26 unique PAMs is available at "CrisPam: SNP-Derived PAM Analysis Tool for Allele-Specific Targeting of Genetic Variants Using CRISPR-Cas Systems" by Rabinowitz et al. Therefore, the use of only 8 PAMs should be reasoned.

3. CRISPR nomenclature: It is better to mention Cpf1 as Cas12a (see the Koonin papers on CRISPR classifications).

4. The Git page does not exist (<https://github.com/lanealone/CRISPR>) (as for Sept 10). Once online, the Git page should include a readme file that describes clearly and thoroughly the use of the scripts.

5. Some of the analyzed Cas enzymes have several PAMs (e.g. xCas9). It might be worth adding an analysis of enrichment per variant (while considering all its potential PAMs).

6. Miscitation of the scientific literature: ref. 1 is irrelevant in the case of describing the CRISPR-Cas system. The proper reference would be:

a. Barrangou, R. et al. CRISPR Provides Acquired Resistance Against Viruses in Prokaryotes. *Science*. 315, 1709–1712 (2007).

7. Miscitation of references 2-4. Gene-editing in plants and drosophila seems irrelevant to support the power of the CRISPR-Cas system in editing human cells. Instead, cite the followings:

a. Jinek, M. et al. A programmable dual-RNA-guided DNA endonuclease in adaptive bacterial immunity. *Science*. 337, 816–21 (2012).

b. Cong, L. et al. Multiplex genome engineering using CRISPR/Cas systems. *Science*. 339, 819–23 (2013).

8. Miscitation of reference 8. Instead cite Jinek et al. ("A programmable dual-RNA-guided DNA endonuclease in adaptive bacterial immunity")

Minor concerns:

9. The figures are well prepared and clearly represent the data.

10. The authors demonstrate that although the %GC of the chromosomes is lower than 50% (sup. fig.

1), there is a clear trend, at least in the traits they tested, for GC enrichment (figs 1&2). It might be worth mentioning in the discussion section.

11. In the discussion section the authors mention that this work was done with the intention to investigate the future potential of facilitating high-throughput CRISPR screening. It would improve the manuscript if the authors elaborate further on how their findings could benefit such future works (discussion) and provide more background on such CRISPR screens (intro).

Reviewer #3 (Remarks to the Author):

The manuscript by Shen and coworkers investigated the human genomic region condensed with different CRISPR PAM (protospacer-adjacent motifs) and evaluated the influenceable region by CRISPR, characterized by eight different PAMs for human complex traits and diseases. The idea is simple and interesting and the analyzed data is useful. Nonetheless, the manuscript suffers from a number of weaknesses, the primary one being that no major leap in understanding is offered and the results are somewhat premature. The current data on the results is not enough and the conclusion is insufficient.

Major concern

1. The authors must experimentally validate the hypothesis that CRISPR editing with three PAMs (NGG, NGAG and NGCG) interferes more with complex human traits and diseases than other PAMs.
2. Although there may be effects of CRISPR/Cas-RNP on the PAM-rich regions of the genome, the off-target effects by the target sequence has much more important. The authors need to consider the off-target effects.
3. Statistically, the sequence of NGG and (NGAG or NGCG) can be found 16 and 4 times more frequently than (TTTA or TTTC or TTTG) on the genome, respectively. So the enrichment of each PAMs cannot be compared with others directly. In Figure 1, the sequence enrichments must be recalculated.

Minor concern

- The description of sequence analysis is lacking. For example, there is no description or expansion about SNP.

Reviewer #4 (Remarks to the Author):

Summary

The present manuscript investigates the suitability of different kinds of CRISPR/Cas protospacer adjacent motifs (PAMs) for the modification of genomic DNA regions associated with the heritability of human diseases and other traits. The study cleverly combines (i) the differential GC content of PAM sequences for different types of Cas molecules with (ii) the established knowledge that higher-GC genomic regions are associated with higher gene content and expression. The manuscript reads well, pursues an interesting premise with wider interest in genetics and CRISPR-based screening assays, and employs up-to-date statistical methods and a range of publicly available data for its meta-analyses and clear findings. On the downside, the abstract is imprecise, state-of-the-art knowledge and availability of Cas molecules with intermediate PAMs are not considered, some methodology was inadvertently not accessible to review owing to an invalid URL, and existing knowledge and work on GC-rich DNA is not cited, so that some novelty aspects of the manuscripts are implicitly overstated. The paper nevertheless already adds significantly to our understanding of regional genomic DNA GC content and heritability of different traits, and given updated analyses has the potential to become an important point of reference for future consideration of different PAM variants for CRISPR screens.

Major Comments

[1]

The very short abstract does not do the study justice, in part because of imprecision in its language, in part because a longer abstract of up to 150 words would be better at conveying the premise and findings of this study. I recommend including more information (e.g. on Cas types, traits investigated and findings).

Regarding imprecisions:

{13-14} The following phrase has two ambiguous word choices that need rephrasing:

"genomic regions condensed with different [...] motifs (PAMs)"

-> "genomic regions enriched with [...] motif (PAM) variants from different types of CRISPR/Cas systems"

{15-16} The second occurrence of "enriched" in the abstract could then be replaced (also for the sake of greater precision) as follows:

"can interfere with the genomic regions that harbour enriched heritability for human complex diseases"

-> "can modify genomic regions that show overrepresentation of sequence elements linked to human complex diseases and other traits"

{16} CRISPR systems do not "capture" DNA, and Cas12a is the official term for Cpf1:

"Cpf1 cannot capture such functional DNA"

-> "such functional DNA is generally inaccessible to [modification by] Cas12a (also known as Cpf1)"

[2]

The introductory section and analyses for Cas9 sites in the genome should consider additional naturally occurring and engineered Cas9 species, including the near-PAMless Cas9 versions designed more recently by Kleinstiver's group (<https://pubmed.ncbi.nlm.nih.gov/32217751/>, with NGN and NRN PAMs) and others. Most relevant in the context of this manuscript, with its focus on traditional DSB-mediated editing, would be those DSB-inducing variants made available via Addgene (<https://www.addgene.org/crispr/cut/>), as they would be at the heart of "facilitating high-throughput CRISPR screening," the future potential of which this manuscript seeks to investigate. The "eight commonly used PAMs" characterised here are in common use because they were published between 2013 and 2018; this might be different going forward.

I therefore request that more and newer literature covering existing variants and PAMs with reasonable cleavage efficiencies (see also my Major Comment [3]) is taken into account in the introductory section, beyond current references 11 and 12, for both Cas9 and Cas12a. I likewise strongly suggest that the analysis of genomic DNA regions suitable for modification by the two platforms is extended at least by including variant Cas12a PAMs with higher GC content (e.g. from here: <https://www.ncbi.nlm.nih.gov/pubmed/28581492>, <https://www.ncbi.nlm.nih.gov/pubmed/30239882>, <https://pubmed.ncbi.nlm.nih.gov/32107556/>). A fuller review and understanding of existing PAMs (see my suggestion for Table S1) would also have told the authors that e.g. GAAG and its reverse complement CTTC (<https://pubmed.ncbi.nlm.nih.gov/32424114/>, <https://pubmed.ncbi.nlm.nih.gov/32876764/> for Cas9 and <https://pubmed.ncbi.nlm.nih.gov/28581492/> as NTTC for Cas12a) and the GAGA reverse complement TCTC (<https://pubmed.ncbi.nlm.nih.gov/28581492/> for Cas12a), used here as non-PAM sequences in Figure 2, are indeed effective PAMs for certain Cas molecules, as are GAGA (<https://pubmed.ncbi.nlm.nih.gov/26474066/> for Cas9) and GTTG (<https://pubmed.ncbi.nlm.nih.gov/33782402/> for Cas12a) with more limited efficiency.

I understand (and appreciate) the smarts of turning a differential distribution of GC content between DNA of known and unknown function into a narrative about different CRISPR platforms, but that narrative must fit the current state of the art for CRISPR application and the tools available. To couch their findings in a suitable biomedical context, the authors should also briefly outline the benefits of using Cas12a (vs Cas9) in research, diagnostic and therapeutic applications. Unwelcome as the request might be, either the analysis has to be extended as indicated, or the narrative, of currently Cas9 vs Cas12a, has to change substantially and the manuscript needs to make its case based on that new angle. Incidentally, the predictable result that new PAM variants with intermediate GC content may have closed the gap between the Cas9 and Cas12a platforms in their utility for analyses and manipulation of disease- (and trait-) related sequences would not be a bad story altogether.

[3]

xCas has NG as a third PAM (<https://pubmed.ncbi.nlm.nih.gov/29512652/>), which has not been analysed here. A very good case should be made for that omission, or the analysis should be included. As a more general note and independent of action taken regarding Major Comment [2], the paper needs to define an efficiency cut-off across all Cas/PAM combinations analysed here, in order to justify the range of PAMs included and excluded from analysis for the shortlisted Cas molecules.

[4]

Give a rationale for excluding gonosomes and in particular the X-chromosome from all analyses, in light of the X-chromosome contributing to many monogenic traits and, by inference, likely contributing to many complex traits. The manuscript emphasises the role of GC content, its correlation with heritable traits and the necessity of choosing suitable PAMs towards improved future CRISPR screens. However, for that precise purpose, inclusion of the X-chromosome in the present analysis would be of great utility.

[5]

Comparison (or is it correlation?) of PAM-featured SNP annotations with other functional annotation is key to the results of this manuscript, and "we compared these annotations with other established functional annotations (see URLs)" does not cover the approach adequately for non-programmers in the readership, even where the URLs are all accessible. Incidentally, the URLs are not formatted as hyperlinks, and it appears that the critical <https://github.com/lanealone/CRISPR> URL is inaccessible, turning the comparison/correlation into a black box for the current round of revisions.

Thinking of the wide readership of Nature Communications, in this context it is also not helpful that Table S2 contains a column heading "URL," which might cause confusion, and that the URLs in question are not included in the Methods section (which to my mind they should) but in a separate URLs section of the authors' own making, if I interpret heading levels correctly by font size.

[6]

The manuscript suggests that a correlation between high-GC gDNA regions and heritability of disease or other traits is a novel finding in this manuscript. While alluding to "previous knowledge" (line 47) concerning higher gene density of chromosomes with higher GC content, no corresponding references are given, and subsequently the authors write that they "hypothesised that GC content has an essential contribution to human complex traits' inheritance" (lines 81-83), concluding that "GC-enriched PAMs have higher potential [of interfering with] human complex traits." The relationship between elevated GC content and coding sequences has been established and elaborated for over thirty years, since the pioneering work by the Bernardi group (<https://pubmed.ncbi.nlm.nih.gov/1908020/>, <https://pubmed.ncbi.nlm.nih.gov/2055469/>, <https://pubmed.ncbi.nlm.nih.gov/8673288/>), including more recent work e.g. on the organisation of GC-rich topologically associated domains away from the nuclear envelope and with close correlation between GC content, transcriptional activity and gene density (<https://pubmed.ncbi.nlm.nih.gov/31601866/>). It being a small step from high gene density and high-

level transcription of genome regions to their likely elevated effect on observable phenotypes, the aforementioned (and other) pre-existing work does not altogether eliminate the novelty of the present findings but qualifies it substantially: a correlation between high-GC genome sequences and both, high-GC PAMs and sequences with elevated heritability. The authors need to present and reference existing work and present their accordingly more moderate hypotheses and findings in that context.

Minor Comments

{20} "element" -> "elements"

{28} "Cpf1" is no longer the standard term and should at least at first occurrence be accompanied by the official term "Cas12a," also in the main text.

{30-31} "Synthetic SpCas9s with altered PAM binding specificity have also been identified^{11,12}." is at once imprecise and too limiting. Synthetic variants with altered PAM binding specificity were created (this was a targeted effort and the sentence suggests that changed specificity was incidental; a slightly petty point, I admit), while natural variants (not mentioned here, which they should) were identified. See also Major Comment [2].

{43}, {104} "144Kb" -> "144 kb" [space-separate value from unit and use minor-case k for 1000x, instead of capital K for 1024x]

{51} "resulted" -> "resulting"

{70} "G/C" -> "GC" [in line with "GC content" and to avoid misinterpretation of "G/C" as a ratio]

{79} "showed" -> "shown"

{80} "In the contrary" -> "On the contrary"/"By contrast"/"Conversely"

{87} "Intended" -> "Intending"

{93} "higher potential in interfering human" -> "higher potential of interfering with human"

{112-113} "the – overlap-annot argument" -> "the --overlap-annot argument" [Suppress Word autoformatting of double hyphens]

{119-120} "28 traits covered [...] attainment, and so on." -> "28 traits included [...] attainment."

{137-177} Correct capitalisation of references throughout. Formatting titles in sentence case should still leave CRISPR, Cas, Cpf1, PAM, Staphylococcus etc with their customary capitalisation.

{Figure 1} The use of 95% interval markers in the Figure is helpful but only qualitative for significance and ambiguous for markers close to enrichment=1. Unfortunately (i) the order of columns and categories does not follow that given in Table S3 (or vice versa), which does not facilitate the identification of p values of specific Trait/PAM combinations in the Figure. Moreover, (ii) NGAG is not covered in Table S3. Both points (i) and (ii) should be rectified.

{Figure 2} Remove additional hidden characters in the manuscript PDF, which interfere with text searches in the document.

{Table S1-3} The description for each Supplementary Table should give abbreviations, sorting criteria and (where appropriate) underlying statistical analyses. All the column headings should be explained (e.g. for Table S3), where these are not self-explanatory.

{Table S1}

A better reference for NGG would be <https://pubmed.ncbi.nlm.nih.gov/23873081/> from 2013.

As IUPAC code is used elsewhere, write "GAA, GAT" -> "GAW (W stands for A and T)"

The title and content of column A seem rather ad hoc, with inconsistent use of italics and the information given, despite the brevity of the table.

With a fuller embrace of the CRISPR PAM theme chosen for this manuscript, and aided by many reviews with overviews of PAM variants, Table S1 could be expanded to the full spectrum of currently published PAMs and references, sorted by PAM GC content and additionally giving (i) a binary Cas9/Cas12a classification to illustrate their distribution within the spectrum, (ii) unambiguous references not present in the manuscript, and (iii) the reason why specific PAMs were included in the analysis. Besides supporting the current article objective (facilitation of future CRISPR screens), such a comprehensive table would make the manuscript a more frequent point of reference.

{Table S2}

"waist circumstance" -> "waist circumference"; "hyperractivity" -> "hyperactivity"

The URL <https://www.thessgac.org/data> requires institutional login for accessibility, and a more basic URL should be given to avoid access errors for readers.

Likewise, the URL https://pgcdata.med.unc.edu/other_gwas_downloads/tag/tag.evrsmk.tbl.gz and its domain do not appear to be accessible; give an accessible URL.

{Table S3}

Legend: "corresponds one" -> "corresponds to one"

The default sort order of this table is unclear and does not match the order of elements in Figure 1. Per PAM there are two alphabetically sorted lists of traits, but the logic for separation of those lists is not apparent. Correspondence of the order of rows (if not also of colours) here with Figure 1 would be helpful to the reader.

Point-to-point responses to reviewers

for “Contribution of CRISPRable DNA on human complex traits” (1st revision)

April 11, 2022

1 Responses to Reviewer #1

1. The general relevance of this study is unclear. It should be indicated right from the abstract – what is the relevance of the results. Currently, it seems that the PAM-annotations don't mean much more than a GC content analysis of human complex traits. The authors should elaborate whether there is an advantage to the enrichment they describe, where most, if not all genomic regions can be targeted by several gRNAs of any Cas variant (i.e. the probability of having a PAM sequence at the region of interest is very high, especially if using PAMless variants – see the next comment).

A: We apologize that the aim or relevance was not conveyed clear enough. This seems to be partly due to our “quantitative genetics” language rather than describing a molecular biology story. In the revision, we have expanded the entire main text, with more detailed introductions, descriptions, and discussions. We emphasize that the aim of this study is to provide a genomic landscape view of human complex traits, with respect to CRISPRable regions of the genome. Regarding relevance, we do agree with the reviewer that eventually the heritability enrichment at PAM-annotations seems to be “simply” caused by enriched GC content. Nevertheless, we did not know this before the analysis. Given the relationship between elevated

GC content and coding sequences in established literature (see also Responses to Reviewer #4, point 6) and the enrichment of complex traits heritability in coding regions, one might infer that GC content itself is driving the observed heritability enrichment. This might make the enrichment of complex traits heritability in GC-enriched regions seem obvious, but no literature systematically analyzed this using the post-GWAS data. In fact, our analysis seems to reveal that heritability enrichment on different “functional genomic annotations” (a hot human quantitative genetics topic) could mostly be simply driven/explained by GC content. Although we report such a conclusion in terms of its relevance to CRISPR, it is also a general conclusion that could be useful in other complex traits genetics studies.

Nevertheless, as the reviewer pointed out, each short PAM can exist almost everywhere along the genome, and a specific PAM does not have to target a GC-enriched piece in the genome. In the revision, we tried to demonstrate the distribution of GC content and the distribution of different types of PAMs (**Figure 2**). Although GC content itself substantially explains the heritability enrichment of complex traits, there can still be a piece of the genome that harbor a high number of a GC-enriched PAM sequence which is not one of the top GC-rich pieces. Our conclusion provides a general landscape that, when a particular PAM is widely applied to high-throughput genome editing, more genetic effects could be altered if the PAM contains more G/C bases. In the practice of CRISPR-Cas techniques, we suggest that one should choose carefully to balance the PAM specificity and GC content, for both high editing coverage and noticeable changes in genetic effects.

We included the above discussions in the revised Discussion (lines 186-208).

2. The limited PAM selection of 8 different PAMs is unclear. The authors should consider mentioning a recent work by Walton et al. (“Unconstrained genome targeting with near-PAMless engineered CRISPR-Cas9 variants”) where the researchers developed Cas variants with a limited PAM requirement. Moreover, a list of 26 unique PAMs is available at “CrisPam: SNP-Derived PAM Analysis Tool for Allele-Specific Targeting of Genetic Variants Using CRISPR-Cas Systems” by Rabinowitz et al. Therefore, the use of only 8 PAMs should be reasoned.

A: In the revision, we expanded our analysis to 77 unique PAM sequences that corresponds to 21 Cas enzymes, including the sixteen Cas9 enzymes, five Cas12a enzymes (See Supplementary Table 1). These Cas enzymes includes both natural Cas enzymes and their engineered variants with altered PAM specificity. In addition, their PAMs vary in length and base composition (e.g., GC content), making them a good representative subgroup of the CRISPR-Cas systems used in genome editing.

In our analysis, we aim to investigate whether regions that are most editable by certain Cas enzyme harbor enriched heritability. The most editable regions are determined by counting and ranking the number of PAMs existed on both strands in each region. So for the SpG variant developed by Walton et al., it's equivalent to the analysis of GC content (the number of G's and C's, and the number of C's matches the number of G's on the reverse strand), which has been included in our analysis (see **Figure 1**). However, the SpRY variant, which requires the NRN (R is A or G) PAM, we cannot determine the most editable regions, as theoretically it can recognize every single base.

3. CRISPR nomenclature: It is better to mention Cfp1 as Cas12a (see the Koonin papers on CRISPR classifications).

A: Thanks for the suggestion. We have updated the nomenclature in revised version of this manuscript.

4. The Git page does not exist (<https://github.com/lanealone/CRISPR>) (as for Sept 10). Once online, the Git page should include a readme file that describes clearly and thoroughly the use of the scripts.

A: We apologize for the outdated GitHub link (due to the change of our GitHub username). The scripts used in our analysis are now available at <https://github.com/RanranZhai/CRISPR>, with an updated readme file.

5. Some of the analyzed Cas enzymes have several PAMs (e.g. xCas9). It might be worth adding an analysis of enrichment per variant (while considering all its potential PAMs).

A: In the revised version of our paper, we have included all its potential PAMs for each Cas enzyme where applicable (see Supplementary Table 1). Note that though e.g. xCas9 has three PAM sequences (NG, GAA, and GAT), we only counted the number of NG PAM in each segment when choosing the Cas-enriched segments, as the count of GAA and GAT PAMs were already covered by the NG PAM.

6. Miscitation of the scientific literature: ref. 1 is irrelevant in the case of describing the CRISPR-Cas system. The proper reference would be: a. Barrangou, R. et al. CRISPR Provides Acquired Resistance Against Viruses in Prokaryotes. *Science*. 315, 1709–1712 (2007).

A: Thanks for the careful review of our citations. We have corrected these in the revision.

7. Miscitation of references 2-4. Gene-editing in plants and drosophila seems irrelevant to support the power of the CRISPR-Cas system in editing human cells. Instead, cite the followings: a. Jinek, M. et al. A programmable dual-RNA-guided DNA endonuclease in adaptive bacterial immunity. *Science*. 337, 816–21 (2012). b. Cong, L. et al. Multiplex genome engineering using CRISPR/Cas systems. *Science*.

339, 819–23 (2013).

A: Corrected.

8. Miscitation of reference 8. Instead cite Jinek et al. (“A programmable dual-RNA-guided DNA endonuclease in adaptive bacterial immunity”)

A: Corrected.

Minor concerns: 9. The figures are well prepared and clearly represent the data.

A: Thanks. During the revision, we spent quite some efforts to update the figures, which contain more results and give a better overview of the conclusion.

10. The authors demonstrate that although the %GC of the chromosomes is lower than 50% (sup. fig. 1), there is a clear trend, at least in the traits they tested, for GC enrichment (figs 1&2). It might be worth mentioning in the discussion section.

A: We apologize if we were not clear enough in the previous version. This is about the difference between the %GC in the genome and the %heritability of a complex trait that the GC-regions capture. This is somewhat related to the first point from the reviewer. Although it seems at the end the heritability enrichment we inferred from the PAMs simply reflects the GC content, this work is still a systematic investigation of GC-vs-AT analysis of complex trait inheritance. The significance is because a small proportion of the genome (GC bases) captures a large proportion of the genetic effects (lines 104-108 in the Results section). In the revision, we expanded the discussion (lines 177-185).

11. In the discussion section the authors mention that this work was done with the intention to investigate the future potential of facilitating high-throughput CRISPR screening. It would improve the manuscript if the authors elaborate further on how their findings could benefit such future works (discussion) and provide more background on such CRISPR screens (intro).

A: For this point, we have consulted an expert in gene editing experiment, Dr. Xiaoyan Jia, ASHG 2021 C.W. Cotterman Awardee, who is acknowledged in the revised manuscript. As far as we understood, the technique so-called “prime editing” aims to conduct CRISPR across the genome in a high-throughput manner. Although the technique is not perfect yet, we can foresee that altering the genetic basis for a complex trait or disease would be possible in the future. In this regard, it is essential to know that PAMs with enriched GC content is much more likely to influence the genetic regulation in general, which is the key message we try to convey in this paper. We added discussions in the revision (lines 220-226) and also some introduction about prime editing in Introduction (lines 33-37).

2 Responses to Reviewer #3

1. The authors must experimentally validate the hypothesis that CRISPR editing with three PAMs (NGG, NGAG and NGCG) interferes more with complex human traits and diseases than other PAMs.

A: We agree with the reviewer that, if possible, experimentally validating CRISPRing the genome with different PAMs would substantially strengthen the story. However, 1) The aim of this study is to provide a genomic landscape view of human complex traits, with respect to CRISPRable pieces of the genome. This can be justified by heritability enrichment of each complex trait on our defined genome pieces, and our conclusion is supported by not only one trait, but also by the genetic architecture of multiple complex phenotypes in different categories; 2) Experimentally validating such results would require high-throughput targeted gene editing across the genome, which is extremely difficult to achieve with sufficient efficiency at present, though the cutting-edge technology on this topic named “prime editing” may have a potential to achieve this in the future; 3) Ethically, we would not be able to validate the exact human complex traits variation in living humans. Cell lines can be used but cannot imply human complex traits. Complex traits in single-cell organisms can be considered, but the relevance to humans is little. For the last two points above, we have consulted an expert in gene editing experiment, Dr. Xiaoyan Jia, ASHG 2021 C.W. Cotterman Awardee, who is acknowledged in the revised manuscript. We think this point cannot be practically addressed at present. Nevertheless, we did spend significant efforts addressing the other comments, and we hope the reviewer finds our revision satisfactory.

2. Although there may be effects of CRISPR/Cas-RNP on the PAM-rich regions of the genome, the off-target effects by the target sequence has much more important. The authors need to consider the off-target effects.

A: We thank the reviewer for raising this common concern. First, the off-target effects are mainly caused by the tolerance of mismatches between the target sequence and single-guide RNA (sgRNA), which is actually beyond the scope of our investigation since we focus on the genomic regions that are enriched or more condensed with PAM sequences rather than specific sgRNAs. Secondly, even though the broadened PAM requirement may increase the propensity for off-targeting, as suggested by Collias et al., the lack of a resource of off-target effects evaluation of different PAMs across the human genome has made it difficult to take the off-target effects into consideration in our current analysis. Although off-target effects cannot be accurately quantified in the analysis, we believe it does not preclude our main conclusion from being true,

i.e., high-throughput CRISPR technique with GC-enriched PAMs would be more likely to alter the genetic effects for complex traits or diseases. We added discussions regarding the off-target effects in the revision (lines 209-219).

3. Statistically, the sequence of NGG and (NGAG or NGCG) can be found 16 and 4 times more frequently than (TTTA or TTTC or TTTG) on the genome, respectively. So the enrichment of each PAMs cannot be compared with others directly. In Figure 1, the sequence enrichments must be recalculated.

A: This is clearly a misunderstanding, and we apologize for any lack of clarity in the previous version. First, the definition of *heritability enrichment* on an annotated piece of genome is “the proportion of heritability captured by this piece divided by the proportion of SNPs in this piece”, so the heritability captured is normalized by the number of SNPs (or length of the genome piece). Second, for each Cas enzyme, we selected the top 2,000 pieces out of the total 20,000 pieces of the genome. Regardless of the PAMs, we always look at the heritability captured by 2,000 equally sized pieces from the genome, so more frequent PAMs would not be over-weighted. The frequencies of PAMs in each piece only affects which 2,000 pieces would be ranked top, but not causing bias in the heritability enrichment analysis. In the revision, we expanded the definition of heritability enrichment in the Results (lines 104-108).

Minor concern

- The description of sequence analysis is lacking. For example, there is no description or expansion about SNP.

A: We apologize for this. The abbreviation is now explained, and we added subsections in the Methods specifically describing how the genome was annotated for particular PAMs (lines 228-250).

3 Responses to Reviewer #4

1. The very short abstract does not do the study justice, in part because of imprecision in its language, in part because a longer abstract of up to 150 words would be better at conveying the premise and findings of this study. I recommend including more information (e.g. on Cas types, traits investigated and findings).

A: We are sorry for the short and imprecise abstract as there is a limitation of up to 150 words when we initially submitted this manuscript to *Nature Biotechnology* as a Brief Communication and later got directly transferred from there. We have now revised the abstract to be more informative.

2. The introductory section and analyses for Cas9 sites in the genome should consider additional naturally occurring and engineered Cas9 species, including the near-PAMless Cas9 versions designed more recently by Kleinstiver's group (<https://pubmed.ncbi.nlm.nih.gov/32217751/>, with NGN and NRN PAMs) and others. Most relevant in the context of this manuscript, with its focus on traditional DSB-mediated editing, would be those DSB-inducing variants made available via Addgene (<https://www.addgene.org/crispr/cut/>), as they would be at the heart of "facilitating high-throughput CRISPR screening," the future potential of which this manuscript seeks to investigate. The "eight commonly used PAMs" characterised here are in common use because they were published between 2013 and 2018; this might be different going forward.

I therefore request that more and newer literature covering existing variants and PAMs with reasonable cleavage efficiencies (see also my Major Comment [3]) is taken into account in the introductory section, beyond current references 11 and 12, for both Cas9 and Cas12a. I likewise strongly suggest that the analysis of genomic DNA regions suitable for modification by the two platforms is extended at least by including variant Cas12a PAMs with higher GC content (e.g. from here: <https://www.ncbi.nlm.nih.gov/pubmed/28581492>, <https://www.ncbi.nlm.nih.gov/pubmed/30239882>, <https://pubmed.ncbi.nlm.nih.gov/32107556/>). A fuller review and understanding of existing PAMs (see my suggestion for Table S1) would also have told the authors that e.g. GAAG and its reverse complement CTTC (<https://pubmed.ncbi.nlm.nih.gov/32424114/>, <https://pubmed.ncbi.nlm.nih.gov/32876764/> for Cas9 and <https://pubmed.ncbi.nlm.nih.gov/28581492/> as NTTC for Cas12a) and the GAGA reverse complement TCTC (<https://pubmed.ncbi.nlm.nih.gov/28581492/> for Cas12a), used here as non-PAM sequences in Figure 2, are indeed effective PAMs for certain Cas molecules, as are GAGA

(<https://pubmed.ncbi.nlm.nih.gov/26474066/> for Cas9) and GTTG (<https://pubmed.ncbi.nlm.nih.gov/33782402/> for Cas12a) with more limited efficiency.

I understand (and appreciate) the smarts of turning a differential distribution of GC content between DNA of known and unknown function into a narrative about different CRISPR platforms, but that narrative must fit the current state of the art for CRISPR application and the tools available. To couch their findings in a suitable biomedical context, the authors should also briefly outline the benefits of using Cas12a (vs Cas9) in research, diagnostic and therapeutic applications. Unwelcome as the request might be, either the analysis has to be extended as indicated, or the narrative, of currently Cas9 vs Cas12a, has to change substantially and the manuscript needs to make its case based on that new angle. Incidentally, the predictable result that new PAM variants with intermediate GC content may have closed the gap between the Cas9 and Cas12a platforms in their utility for analyses and manipulation of disease- (and trait-) related sequences would not be a bad story altogether.

A: We thank the reviewer for providing such a detailed list of recent CRISPR development literature. In the revision, we expanded our analysis to 77 unique PAM sequences that corresponds to 21 Cas enzymes, including sixteen Cas9 enzymes and five Cas12a enzymes (Supplementary Table 1, **Figure 1**). These Cas enzymes includes both natural Cas enzymes and their engineered variants with altered PAM specificity. The updated list of Cas enzymes includes the AsCas12a RR variant, which uses PAMs with higher GC content, so indeed not only the Cas9 enzymes can target genome regions that harbor more heritability of complex traits. The results now show a more clear picture about the strong relationship between the GC content of PAMs and heritability enrichment of human complex traits.

3. xCas has NG as a third PAM (<https://pubmed.ncbi.nlm.nih.gov/29512652/>), which has not been analysed here. A very good case should be made for that omission, or the analysis should be included. As a more general note and independent of action taken regarding Major Comment [2], the paper needs to define an efficiency cut-off across all Cas/PAM combinations analysed here, in order to justify the range of PAMs included and excluded from analysis for the shortlisted Cas molecules.

A: In the revision, we have included the NG PAM of xCas9, which has made the xCas9 equivalent to the SpCas9-NG, as the NG PAM covers the other PAMs (GAW; W is for A and T) of xCas9 in our analysis. The NG PAM annotation in fact directly reflects the GC content across the genome.

4. Give a rationale for excluding gonosomes and in particular the X-chromosome from all analyses, in light of the X-chromosome contributing to many monogenic traits and, by inference, likely contributing to many complex traits. The manuscript emphasises the role of GC content, its correlation with heritable traits and the necessity of choosing suitable PAMs towards improved future CRISPR screens. However, for that precise purpose, inclusion of the X-chromosome in the present analysis would be of great utility.

A: Chromosome X is certainly as important (or sometimes even more important) as the autosomes. Our group is also working on statistical methods incorporating the X chromosome into standard genomic analysis. At present, the S-LDSC tool we used to assess heritability enrichment cannot include the X chromosome, as no X chromosome functional annotations are available for the analysis. To address this point, we have tried our best to compile a list of functional annotations (e.g., coding regions, H3K27 methylation, and enhancer) on the X chromosome (**Figure 4**). We found a similar pattern of overlapping with GC content and GC-enriched PAMs for these functional annotations. This suggests that the conclusion for the autosomes also applies to the X chromosome, as long as the molecular tools for gene editing do not perform very differently for the X chromosome.

5. Comparison (or is it correlation?) of PAM-featured SNP annotations with other functional annotation is key to the results of this manuscript, and “we compared these annotations with other established functional annotations (see URLs)” does not cover the approach adequately for non-programmers in the readership, even where the URLs are all accessible. Incidentally, the URLs are not formatted as hyperlinks, and it appears that the critical <https://github.com/lanealone/CRISPR> URL is inaccessible, turning the comparison/correlation into a black box for the current round of revisions.

Thinking of the wide readership of Nature Communications, in this context it is also not helpful that Table S2 contains a column heading “URL,” which might cause confusion, and that the URLs in question are not included in the Methods section (which to my mind they should) but in a separate URLs section of the authors’ own making, if I interpret heading levels correctly by font size.

A: We apologize for the outdated GitHub link (due to the change of our GitHub username). The scripts used in our analysis are now available at <https://github.com/RanranZhai/CRISPR>, with an updated readme file.

In the sentence “we compared these annotations with other established functional annotations (see URLs)”, we meant to provide URLs for the functional annotations we used. For the comparison (could be interpreted as correlation) between our annotations and the established functional annotations, we used the

Fisher's exact test to evaluate statistical significance. We clarified this in the revised Figure legends and also tried to improve the layout of Supplementary Table 2 for clarity.

A minor note: This manuscript was transferred directly from *Nature Biotechnology* to *Communications Biology* (instead of *Nature Communications*). Nevertheless, we hope that publications in this new open access journal will reach as wide readership as the other Nature-branded journals.

6. The manuscript suggests that a correlation between high-GC gDNA regions and heritability of disease or other traits is a novel finding in this manuscript. While alluding to "previous knowledge" (line 47) concerning higher gene density of chromosomes with higher GC content, no corresponding references are given, and subsequently the authors write that they "hypothesised that GC content has an essential contribution to human complex traits' inheritance" (lines 81-83), concluding that "GC-enriched PAMs have higher potential [of interfering with] human complex traits." The relationship between elevated GC content and coding sequences has been established and elaborated for over thirty years, since the pioneering work by the Bernardi group (<https://pubmed.ncbi.nlm.nih.gov/1908020/>, <https://pubmed.ncbi.nlm.nih.gov/2055469/>, <https://pubmed.ncbi.nlm.nih.gov/8673288/>), including more recent work e.g. on the organisation of GC-rich topologically associated domains away from the nuclear envelope and with close correlation between GC content, transcriptional activity and gene density (<https://pubmed.ncbi.nlm.nih.gov/31601866/>). It being a small step from high gene density and high-level transcription of genome regions to their likely elevated effect on observable phenotypes, the aforementioned (and other) pre-existing work does not altogether eliminate the novelty of the present findings but qualifies it substantially: a correlation between high-GC genome sequences and both, high-GC PAMs and sequences with elevated heritability. The authors need to present and reference existing work and present their accordingly more moderate hypotheses and findings in that context.

A: We again thank the reviewer for pointing us to important literature. We apologize for not clearly demonstrating the novelty in this regard, and we had no intention to ignore the established literature on "the relationship between elevated GC content and coding sequences". We would like to also note here that although eventually the heritability enrichment at PAM-annotations seems to be "simply" caused by enriched GC content, we did not know this before the analysis. Given "the relationship between elevated GC content and coding sequences" and the enrichment of complex traits heritability in coding regions, one might infer that GC content itself is driving the observed heritability enrichment. This might make the enrichment of complex traits heritability in GC-enriched regions seem obvious, but no literature system-

atically analyzed this using the post-GWAS data. In fact, our analysis seems to reveal that heritability enrichment on different “functional genomic annotations” (a hot human quantitative genetics topic) could mostly be driven/explained by GC content. Although we report such a conclusion in terms of its relevance to CRISPR, it is also a general conclusion that could be useful in other complex traits genetics studies. In the revision, we expanded the discussions on this point (lines 186-208).

Minor Comments

20 “element” -> “elements”

28 “Cpf1” is no longer the standard term and should at least at first occurrence be accompanied by the official term “Cas12a,” also in the main text.

30-31 “Synthetic SpCas9s with altered PAM binding specificity have also been identified^{11,12}.” is at once imprecise and too limiting. Synthetic variants with altered PAM binding specificity were created (this was a targeted effort and the sentence suggests that changed specificity was incidental; a slightly petty point, I admit), while natural variants (not mentioned here, which they should) were identified. See also Major Comment [2].

43, 104 “144Kb” -> “144 kb” [space-separate value from unit and use minor-case k for 1000x, instead of capital K for 1024x]

51 “resulted” -> “resulting”

70 “G/C” -> “GC” [in line with “GC content” and to avoid misinterpretation of “G/C” as a ratio]

79 “showed” -> “shown”

80 “In the contrary” -> “On the contrary”/“By contrast”/“Conversely”

87 “Intended” -> “Intending”

93 “higher potential in interfering human” -> “higher potential of interfering with human”

112-113 “the – overlap-annot argument” -> “the –overlap-annot argument” [Suppress Word autoformatting of double hyphens]

119-120 “28 traits covered [...] attainment, and so on.” -> “28 traits included [...] attainment.”

137-177 Correct capitalisation of references throughout. Formatting titles in sentence case should still leave CRISPR, Cas, Cpf1, PAM, Staphylococcus etc with their customary capitalisation.

A: Again, we thank the reviewer for the great and careful efforts on checking all these details. We have now resolved these issues where applicable in the revised manuscript.

Figure 1 The use of 95% interval markers in the Figure is helpful but only qualitative for significance and ambiguous for markers close to enrichment=1. Unfortunately (i) the order of columns and categories does not follow that given in Table S3 (or vice versa), which does not facilitate the identification of p values of specific Trait/PAM combinations in the Figure. Moreover, (ii) NGAG is not covered in Table S3. Both points (i) and (ii) should be rectified. Figure 2 Remove additional hidden characters in the manuscript PDF, which interfere with text searches in the document.

A: We now have **Supplementary Table 3** aligned with the heatmap in **Figure 1**. The legend of **Supplementary Table 3** in the Supplementary Information provides detailed descriptions. Hidden characters were removed from **Figure 3** (originally **Figure 2**).

Table S1-3 The description for each Supplementary Table should give abbreviations, sorting criteria and (where appropriate) underlying statistical analyses. All the column headings should be explained (e.g. for Table S3), where these are not self-explanatory.

Table S1

A better reference for NGG would be <https://pubmed.ncbi.nlm.nih.gov/23873081/> from 2013.

As IUPAC code is used elsewhere, write “GAA, GAT” -> “GAW (W stands for A and T)”

The title and content of column A seem rather ad hoc, with inconsistent use of italics and the information given, despite the brevity of the table.

With a fuller embrace of the CRISPR PAM theme chosen for this manuscript, and aided by many reviews with overviews of PAM variants, Table S1 could be expanded to the full spectrum of currently published PAMs and references, sorted by PAM GC content and additionally giving (i) a binary Cas9/Cas12a classification to illustrate their distribution within the spectrum, (ii) unambiguous references not present in the manuscript, and (iii) the reason why specific PAMs were included in the analysis. Besides supporting the current article objective (facilitation of future CRISPR screens), such a comprehensive table would make the manuscript a more frequent point of reference.

A: With the expanded Cas enzyme list, we have updated **Supplementary Table 1** in the **Supplementary Tables** file, with the legend given in the **Supplementary Information**.

Table S2

“waist circumstance” -> “waist circumference”; “hyperractivity” -> “hyperactivity”

The URL <https://www.thessgac.org/data> requires institutional login for accessibility, and a more

basic URL should be given to avoid access errors for readers.

Likewise, the URL https://pgcdata.med.unc.edu/other_gwas_downloads/tag/tag.evrsmk.tbl.gz and its domain do not appear to be accessible; give an accessible URL.

A: We really apologize for the typos – clearly in the original version we did not check the grammar carefully for the supplementary materials. We have corrected these and replaced the inaccessible URL with their corresponding publications.

Table S3

Legend: “corresponds one” -> “corresponds to one”

The default sort order of this table is unclear and does not match the order of elements in Figure 1. Per PAM there are two alphabetically sorted lists of traits, but the logic for separation of those lists is not apparent. Correspondence of the order of rows (if not also of colours) here with Figure 1 would be helpful to the reader.

A: We have updated Supplementary Table 3, now ordered by the Cas and then by the trait, following the orders in **Figure 1**. We also provided the false discovery rate (the FDR column) based on the enrichment *P* value (the Enrichment_p column).

Prof. Xia Shen, on behalf of the co-authors

Reviewers' comments:

Reviewer #1 (Remarks to the Author):

Comments to the revised manuscript "Contribution of CRISPRable DNA on human complex traits"

The revised version has been greatly improved. The intro and discussion provide more thorough background and attempt to describe the significance of this work. However, some of the comments were not addressed. The revisions brought up some more comments as described below.

1. In line 29: the proper citation was not integrated although being suggested during the first round of revision. The authors cite Jinek et al alone, without Le Cong et al. The basis for my suggestion is that Jinek et al. first described the use of the CRISPR-Cas system for DNA editing in a programmable manner, while Le Cong et al. showed the use of the system in human cells for the first time.

Therefore, as I previously suggested, the proper citation would be both references.

2. The mention of Prime Editing seems irrelevant. If the authors believe it improves the background section, it seems that other CRISPR-related techniques are missing (such as base editing screening. see works by John Doench). I would not suggest to add any of these (base editing and prime editing) as it does not seem to broaden the relevant background.

3. Line 64: rephrase "These enzymes altogether have enabled us to edit a large proportion of the human genome via CRISPR" as it gives a false impression that this work contains "wet-lab" experiments.

4. While the authors write in the rebuttals letter that they have consulted with an expert in the field and revised the manuscript according to the revisions, the revised manuscript still lacks the basic explanation of CRISPR screenings. It is critical to emphasize that prime editing is not the point, rather another CRISPR technique that can also be used in screening.

5. Line 98: "Thus, in the subsequent analysis, we determined the most editable regions of each Cas by summing the number of all its potential PAMs within each genome segment and denoting the top 2,000 segments (10% of the genome) as Cas enriched regions." The addition of the different PAMs and Cas enzymes is good. Please elaborate, when writing "by summing the number of all its potential PAMs within each genome segment" if that means that for a particular Cas, e.g. VQR SpCas9, the appearance of GAG in the genome would count as two different PAMs – GA and GAG. Thus, the same region counts as 2 or possibly even more PAMs.

6. Discussion "GC-rich regions are enriched with genes, transcription start site (TSS), promoters..." – the abbreviation of TSS should appear earlier as it first appears.

7. I do agree with the following statement: "Our conclusion provides a general landscape that, when a particular PAM is widely applied to high-throughput genome editing, more genetic effects could be altered if the PAM contains more G/C bases". However, I was not convinced that the results, as described, demonstrate meaningful data with biological implications to CRISPR screens. I would not argue if the authors claimed that due to the commonly known property that coding regions are GC rich, such PAMs can be more common in relevant regions. Such claim is obvious but important for CRISPR screens. However, the perspective of the manuscript seems to be confusing or misleading, aiming to demonstrate the CRISPRable regions, rather than just GC content along the genome.

8. The following claim is problematic: "Although systematic validations of such gene editing effects on human complex traits are practically and ethically impossible". Validations can be done on patient derived cells (iPSCs etc) to investigate the mechanism of action of diseases.

9. Although the manuscript has been significantly improved, it still lacks novelty and biological relevance due to the above comments.

Reviewer #2 (Remarks to the Author):

In contrast to the previous version of the manuscript, the revised version is more precise in the abstract and more clearly stated in the manuscript. Also, the authors modified the figure to be clearer

and more correct. In addition to that, I have some minor concerns.

1. As the authors responded, it is difficult to experimentally validate such results. However, the off-target effect is included in the interferences of Cas protein to the whole genome DNA. There are some papers about the collection of CRISPR/Cas off-target effect, such as works by Florian Störtz and Peter Minary (<https://doi.org/10.1093/nar/gkaa885>). I highly recommend the authors include the comparison of off-target frequency in the genomic region and your results, in the discussion section.
2. The description of graphs in figures is lacking, such as axis titles or labels.
3. (line 172) The phrase "21 Cas enriched genomic regions" can be confusing. I think it is better to modify it with "the PAM-enriched genomic regions of 21 Cas".

Reviewer #3 (Remarks to the Author):

Summary

The authors have greatly improved the manuscript overall and have gone to great lengths, including substantial additional text, references and analyses, to address points raised during first-round review, such as inclusion of additional and intermediate PAMs, discussion and/or analysis of enrichment for the X-chromosome and additional points of discussion. Concept, execution, presentation and novelty of findings fully justify publication.

Major Comments

An efficiency cut-off for inclusion of PAMs has not been defined. If this is difficult (given e.g. different means of reporting efficiency across different studies and with the current study itself naturally not including wet work across all PAMs reported), then at least a statement (true to the source literature) to the effect of "Only PAMs explicitly pointed out as efficient in the original reports were included for our meta-analyses in the present study." should be included.

[138-140] it is noteworthy that the top 2,000 PAM-enriched regions were not necessarily the top 2,000 GC-rich regions, even for PAMs that have a relatively high GC content -> it is therefore noteworthy that other factors must contribute to heritability and that increasing correspondence with increasing PAM GC content of the top 2,000 PAM-enriched regions with the top 2,000 GC-rich regions was not absolute and only a trend even for PAMs with top GC content [The logic of this sentence was not clear, and I hope that this suggestion captures its meaning better; otherwise, please, rephrase. This point might then also be made statistically across all PAMs analysed, where e.g. the r for a basic correlation analysis of % (top-2000 PAM regions/top-GC-rich regions) vs. % (PAM GC content) might give a reportable number for the apparent trend or correlation, although the % (PAM GC content) for 2- to 5-nucleotide sequences (of 0/20/25/33/40/50/60/67/75/100%) will be highly granular. On that note, the argument can be made for section lines 238-250, that *internal* N ambiguities for PAMs should be counted as 50% GC content, so that e.g. NGG would have $2/2=100\%$ GC content, but e.g. NGGNG would have $3.5/4=87.5\%$ GC content.]

Minor Comments

[29] Due to its -> Due to their [In this sentence, the plural is more appropriate, although this might be read as a reference back to the previous sentence, where "CRISPR-Cas" is used in the singular.]
[36] functional assaying hundreds -> [either] functionally assaying hundreds [or] functional assaying of hundreds
[38-42] [Use comma-separated "which" twice instead of "that" for what are non-defining relative clauses.]

[40] thus -> so that [Or separate thus with a semicolon instead of a comma.]
[44] double-stranded break -> double-strand break
[45, 46] DSB -> DSBs [Use -s to indicate the plural of acronyms, or consistently avoid doing so elsewhere, e.g. for "PAMs" in lines 20 and 48 or "SNPs" in line 69.]
[49] down-stream -> downstream
[51] require -> requires
[71] complex traits biology -> complex trait biology [as a rule, compound adjectives in the singular]
[74, 189, 191] complex traits heritability -> complex trait heritability [as a rule, compound adjectives in the singular]
[96] PAM enriched -> PAM-enriched
[110] SpCas9 enriched -> SpCas9-enriched
[143] despite that GGCG has a GC content -> [either] despite that GGCG having a GC content [or] although that GGCG has a GC content
[180-181] are more accessible to GC-rich regions, as GC-rich regions are enriched with genes -> have greater access to GC-rich regions, which are in turn enriched for genes [The expression "are accessible to" was not correct in the context, and the "as" created a non sequitur.]
[186-203] [These new sections do not yet read well, and sentences and arguments should be linked better, and wordiness and redundancies removed. At least the following change should be introduced, or something to that effect:]
The heritability enrichment at PAM annotations might seem to be simply driven by the PAM GC content. Nevertheless, we did not know this before the analysis. -> Heritability enrichment for high-GC PAMs, such as the initially characterised SpCas9 NGG PAM, therefore closely aligns with known feature enrichment for high-GC genome sequences {refs 38, 45-47}, although such prior knowledge did not enter our statistical analyses, with its focus on heritability of complex traits.
[196] its -> their
[216] effects evaluation -> effect evaluation
[218] mining -> exploitation [or] utilization
[238, 241] Cas enriched -> Cas-enriched
[241] 66% -> 67% [or] 66.7%
[242] PAM within-> PAMs within
[433] Cas enriched -> Cas-enriched
[438] indicates false discovery rate < 0.05 -> indicates significance after multiple testing correction with a false discovery rate of 0.05
[442, also for all Supplementary Figure legends 2-77] by the GC content (grey) decreasingly -> by decreasing GC content (grey)

[Supplementary Tables – Legends]

ST1

we referred to determine -> we referred to in order to determine [or] we referred to for determination of [or] that provided

ST3

P value (the Enrichment_p column) and the false discovery rate (the FDR column) -> P value (the Enrichment p column) and the P value after false discovery rate correction (the FDR column)

ST4

that are overlapped with -> that overlap with
referring the number -> referring to the number

SF1

was count from the GRCh37 -> was based on the GRCh37

Point-to-point responses to reviewers

for “Contribution of CRISPRable DNA on human complex traits” (2nd revision)

Jul 5, 2022

Contents

1 Responses to Reviewer #1	1
2 Responses to Reviewer #2	5
3 Responses to Reviewer #3	6

1 Responses to Reviewer #1

1. In line 29: the proper citation was not integrated although being suggested during the first round of revision. The authors cite Jinek et al alone, without Le Cong et al. The basis for my suggestion is that Jinek et al. first described the use of the CRISPR-Cas system for DNA editing in a programmable manner, while Le Cong et al. showed the use of the system in human cells for the first time. Therefore, as I previously suggested, the proper citation would be both references.

A: We apologize for missing the reference. This is now properly cited in the revision.

2. The mention of Prime Editing seems irrelevant. If the authors believe it improves the background section, it seems that other CRISPR-related techniques are missing (such as base editing screening. see works by John Doench). I would not suggest to add any of these (base editing and prime editing) as it does not seem to broaden the relevant background.

A: There seems to be a misunderstanding. We introduce prime editing in order to lead to the introduction of saturated prime editing (SPE) – a way toward high-throughout prime editing. This is certainly relevant to the point in our paper: The heritability of human complex traits and diseases are widespread

across the genome; therefore, manipulation of the genetic basis of a complex disease would require a high-throughput gene-editing technique. See also the last paragraph of Discussion (lines 228-234).

3. Line 64: rephrase “These enzymes altogether have enabled us to edit a large proportion of the human genome via CRISPR” as it gives a false impression that this work contains “wet-lab” experiments.

A: We are sorry for the confusion. We have clarified this by using “researchers” instead of “us”.

4. While the authors write in the rebuttals letter that they have consulted with an expert in the field and revised the manuscript according to the revisions, the revised manuscript still lacks the basic explanation of CRISPR screenings. It is critical to emphasize that prime editing is not the point, rather another CRISPR technique that can also be used in screening.

A: Similar to our response to point 2 above, we are not saying that prime editing is the point, but rather SPE could be relevant. Regarding “basic explanation of CRISPR screening”, we believe we have introduced the essential background about CRISPR relevant to this particular study. In general, the results are not about the technique but rather about the potential influence of the edited sites on human complex traits.

5. Line 98: “Thus, in the subsequent analysis, we determined the most editable regions of each Cas by summing the number of all its potential PAMs within each genome segment and denoting the top 2,000 segments (10% of the genome) as Cas enriched regions.” The addition of the different PAMs and Cas enzymes is good. Please elaborate, when writing “by summing the number of all its potential PAMs within each genome segment” if that means that for a particular Cas, e.g. VQR SpCas9, the appearance of GAG in the genome would count as two different PAMs – GA and GAG. Thus, the same region counts as 2 or possibly even more PAMs.

A: For VQR-SpCas9, the appearance of GAG in the genome should not count as two different PAMs. Because the count of GA is the sum of GAA, GAC, GAG, and GAT, where the GAG has been counted. This is also applicable to another Cas9 variant, xCas9, where we only count the number of NG, instead of summing the number of NG, GAA, and GAT. We revised this piece of text by adding a concise clarification “(removing duplications between PAMs)” (line 100).

In the previous version of our analysis, we summed the number of GA and GNG, which was a mistake. We have resolved the problem by excluding the count of GAG when summing the number of all its PAMs; figures and tables are now updated accordingly. In Supplementary Table 1, we colored the PAMs that were excluded from the analysis yellow for clarification.

6. Discussion “GC-rich regions are enriched with genes, transcription start site (TSS), promoters...” – the abbreviation of TSS should appear earlier as it first appears.

A: Corrected. Thanks.

7. I do agree with the following statement: “Our conclusion provides a general landscape that, when a particular PAM is widely applied to high-throughput genome editing, more genetic effects could be altered if the PAM contains more G/C bases”. However, I was not convinced that the results, as described, demonstrate meaningful data with biological implications to CRISPR screens. I would not argue if the authors claimed that due to the commonly known property that coding regions are GC rich, such PAMs can be more common in relevant regions. Such claim is obvious but important for CRISPR screens. However, the perspective of the manuscript seems to be confusing or misleading, aiming to demonstrate the CRISPRable regions, rather than just GC content along the genome.

A: We feel unfortunate that our first revision could not convey this point well enough to the reviewer.

First, we explained in the previous revision that the conclusion might seem obvious *after* we explained the reason, but it doesn’t mean it is already obvious to everyone before seeing our report. As the reviewer says, the conclusion is important for CRISPR screens, which is why we spent effort and wrote this manuscript.

Second, even if we find it obvious that GC-rich PAMs would target GC-rich regions in the genome, we would still be uncertain about what CRISPR based on GC-rich PAMs can do to our complex phenotypes and what kind of complex phenotypes it would affect. There is a gap between biotech researchers in the gene-editing field and human population geneticists, and we showed that knowledge from the two areas together could bring useful findings.

Third, please note that PAM-enriched regions, GC-rich regions, and, e.g., coding regions are *not identical* pieces of the genome. Although they overlap, how much the heritability of complex traits is enriched in GC-rich regions in the genome was not analyzed before, nor in PAM-enriched regions. Starting the analysis by defining GC-rich regions would be different from starting from PAM-enriched regions. The way we constructed the manuscript followed the way we approached the research question. This is reasonable as the conclusion is more relevant to CRISPR scientists than to population geneticists.

With these, we believe the finding is not lacking novelty, and the story is clearly described.

8. The following claim is problematic: “Although systematic validations of such gene editing effects on human complex traits are practically and ethically impossible”. Validations can be done on patient derived cells (iPSCs etc) to investigate the mechanism of action of diseases.

A: First, we certainly agree with the reviewer that some experiments can be done using cell lines, which is exactly what we have already written as the second half of this statement: “the results can provide useful insights in specific research using human cell lines and even in the other species when studying complex phenotypes”.

Second, as we wrote above, this point again shows “the gap between biotech researchers in the gene-editing field and human population geneticists”. We agree with the reviewer that cell lines can be used to “investigate the mechanism of action of diseases”, but not to validate the *effect size* of a genetic variant on human complex diseases. For instance, we cannot experimentally validate how much more cancer risk a person with edited genotypes has compared to a regular person (in terms of *odds ratio*).

In order to be more specific, we further rephrased “gene editing effects on human complex traits” as “gene editing effects on complex traits in a human population” (line 232).

9. Although the manuscript has been significantly improved, it still lacks novelty and biological relevance due to the above comments.

A: See our reply to point 7 above for justifications. Basically, we have to disagree here, particularly on novelty, unless there is literature that has already scientifically claimed our conclusion.

2 Responses to Reviewer #2

1. As the authors responded, it is difficult to experimentally validate such results. However, the off-target effect is included in the interferences of Cas protein to the whole genome DNA. There are some papers about the collection of CRISPR/Cas off-target effect, such as works by Florian Störtz and Peter Minary (<https://doi.org/10.1093/nar/gkaa885>). I highly recommend the authors include the comparison of off-target frequency in the genomic region and your results, in the discussion section.

A: We thank the reviewer for this suggestion, and we spent quite some time investigating the datasets that may potentially be integrated with our analysis. We acquired the data from the crisprSQL database developed by Florian Störtz. After filtering out non-human genome targets, we obtained 9,048 targets from 13 studies, of which 8,956 are off-target sites of SpCas9. We then matched these 8,956 off-target sites to our defined 20,000 genome segments. We found that the more NGG that a segment has, the more off-target sites are within that segment (Supplementary Fig. 78), which fits the theory that PAM-rich regions have an increased propensity for off-targeting. However, the cleavage frequency is poorly correlated with the number of NGG of the corresponding segments, suggesting that the cleavage frequency is not solely dependent on the presence of PAM (NGG). We expanded such discussions in the revision (lines 211-227).

2. The description of graphs in figures is lacking, such as axis titles or labels.

A: We rechecked the figure legends and thought this point was about the lack of axis labels in the subfigures of Figure 2. We have added the axis labels. Thanks for pointing this out.

3. (line 172) The phrase "21 Cas enriched genomic regions" can be confusing. I think it is better to modify it with "the PAM-enriched genomic regions of 21 Cas".

A: Revised accordingly. Thanks.

3 Responses to Reviewer #3

An efficiency cut-off for inclusion of PAMs has not been defined. If this is difficult (given e.g. different means of reporting efficiency across different studies and with the current study itself naturally not including wet work across all PAMs reported), then at least a statement (true to the source literature) to the effect of “Only PAMs explicitly pointed out as efficient in the original reports were included for our meta-analyses in the present study.” should be included.

A: Thank you for your kind suggestion. It is true that the PAMs we used in the study are from many reports, and their measurement of PAM efficiency varies so that a cut-off cannot be well defined. As you suggested, we now added the statement “Only PAMs explicitly pointed out as efficient in the original reports were included in our analysis” (lines 243-244).

[138-140] it is noteworthy that the top 2,000 PAM-enriched regions were not necessarily the top 2,000 GC-rich regions, even for PAMs that have a relatively high GC content -> it is therefore noteworthy that other factors must contribute to heritability and that increasing correspondence with increasing PAM GC content of the top 2,000 PAM-enriched regions with the top 2,000 GC-rich regions was not absolute and only a trend even for PAMs with top GC content [The logic of this sentence was not clear, and I hope that this suggestion captures its meaning better; otherwise, please, rephrase. This point might then also be made statistically across all PAMs analysed, where e.g. the r for a basic correlation analysis of % (top-2000 PAM regions/top-GC-rich regions) vs. % (PAM GC content) might give a reportable number for the apparent trend or correlation, although the % (PAM GC content) for 2- to 5-nucleotide sequences (of 0/20/25/33/40/50/60/67/75/100%) will be highly granular. On that note, the argument can be made for section lines 238-250, that *internal* N ambiguities for PAMs should be counted as 50% GC content, so that e.g. NGG would have $2/2=100\%$ GC content, but e.g. NGGNG would have $3.5/4=87.5\%$ GC content.]

A: We have rephrased in lines 137-144. In this paragraph, we meant to point out that even though there is a trend that enriched segments for PAMs with high GC content are also the top GC-rich segments, there are exceptions. Moreover, these exceptions should be considered when conducting high-throughput genome-editing experiments, which are further discussed in lines 200-210.

We analyzed the correlation between the number of PAM-enriched segments that are also top GC-rich segments and their GC content (Pearson’s $r = 0.86$, $P < 2.2 \times 10^{-16}$) (lines 137-140).

We agree that NGGNG would have $3.5/4=87.5\%$ GC content, which was already shown in Supplementary Table 1 as 88% (rows 53-56).

Minor Comments

- [29] Due to its -> Due to their [In this sentence, the plural is more appropriate, although this might be read as a reference back to the previous sentence, where “CRISPR-Cas” is used in the singular.]
- [36] functional assaying hundreds -> [either] functionally assaying hundreds [or] functional assaying of hundreds
- [38-42] [Use comma-separated “which” twice instead of “that” for what are non-defining relative clauses.]
- [40] thus -> so that [Or separate thus with a semicolon instead of a comma.]
- [44] double-stranded break -> double-strand break
- [45, 46] DSB -> DSBs [Use -s to indicate the plural of acronyms, or consistently avoid doing so elsewhere, e.g. for “PAMs” in lines 20 and 48 or “SNPs” in line 69.]
- [49] down-stream -> downstream
- [51] require -> requires
- [71] complex traits biology -> complex trait biology [as a rule, compound adjectives in the singular]
- [74, 189, 191] complex traits heritability -> complex trait heritability [as a rule, compound adjectives in the singular]
- [96] PAM enriched -> PAM-enriched
- [110] SpCas9 enriched -> SpCas9-enriched
- [143] despite that GGCG has a GC content -> [either] despite that GGCG having a GC content [or] although that GGCG has a GC content
- [180-181] are more accessible to GC-rich regions, as GC-rich regions are enriched with genes -> have greater access to GC-rich regions, which are in turn enriched for genes [The expression “are accessible to” was not correct in the context, and the “as” created a non sequitur.]
- [186-203] [These new sections do not yet read well, and sentences and arguments should be linked better, and wordiness and redundancies removed. At least the following change should be introduced, or something to that effect:] The heritability enrichment at PAM annotations might seem to be simply driven by the PAM GC content. Nevertheless, we did not know this before the analysis. -> Heritability enrichment for high-GC PAMs, such as the initially characterised SpCas9 NGG PAM, therefore closely aligns with known feature enrichment for high-GC genome sequences refs 38, 45-47, although such prior

knowledge did not enter our statistical analyses, with its focus on heritability of complex traits.

[196] its -> their

[216] effects evaluation -> effect evaluation

[218] mining -> exploitation [or] utilization

[238, 241] Cas enriched -> Cas-enriched

[241] 66% -> 67% [or] 66.7%

[242] PAM within-> PAMs within

[433] Cas enriched -> Cas-enriched

[438] indicates false discovery rate < 0.05 -> indicates significance after multiple testing correction with a false discovery rate of 0.05

[442, also for all Supplementary Figure legends 2-77] by the GC content (grey) decreasingly -> by decreasing GC content (grey)

[Supplementary Tables – Legends]

ST1

we referred to determine -> we referred to in order to determine [or] we referred to for determination of [or] that provided

ST3

P value (the *Enrichment_p* column) and the false discovery rate (the FDR column) -> P value (the Enrichment p column) and the P value after false discovery rate correction (the FDR column)

ST4

that are overlapped with -> that overlap with

referring the number -> referring to the number

SF1

was count from the GRCh37 -> was based on the GRCh37

A: All checked and revised. Thanks again for providing such a detailed list of comments!

Prof. Xia Shen, on behalf of the co-authors

REVIEWERS' COMMENTS:

Reviewer #1 (Remarks to the Author):

I have no further comments.

Reviewer #2 (Remarks to the Author):

I generally support the acceptance of this revised manuscript.

Reviewer #3 (Remarks to the Author):

In this revised version of the present manuscript, first- and second-round comments have almost all been addressed in full, and the general conclusions and comprehensive analyses of different PAMs in their correlation with GC-rich regions, now with suitably qualifying statements and numerous language improvements, make this an interesting study, which is citable for its general association of PAM and genome GC content and for a comprehensive look across different widely used PAM sequences.

[Minor points]

[Supplementary Figure 2 to 77]

The current correlation analysis as stated in lines 138 to 140 describes the behaviour across all PAMs. In line with this and going through the supplementary "PAM content vs GC%" plots one more time, it strikes me that an "r" for a simple correlation analysis per PAM would likewise be highly informative for each PAM and more so than enumerating the "number of top-2000 PAM segments that are not in the top-2000 GC-rich segments." By comparison, the present enumeration of non-corresponding top-2000 segments, while informative, feels somewhat construed. More importantly, eyeballing over the graphs, I would naively assume that a correlation of PAM content vs GC% (for the 18,763 segments) would e.g. results in an r approaching 1 for the "GG" PAM and would result in an r approaching -1 for the "AAA" PAM. The r could then be taken as an approximate single measure for a given PAM of how suitable it would be for the targeting of GC rich (and thus by inference "trait-relevant") sections of the genome.

At this point it would be churlish to insist on this analysis, but if the authors agree that r as a measure of PAM content vs GC% per segment would make interpretation of their data more intuitive (and thus more citable), I invite them to add this number to each alignment to strengthen the article.

[Title]

Contribution of CRISPRable DNA on human complex traits -> Contribution of CRISPRable DNA to human complex traits [The preposition "on" is really only used for content contributions e.g. for discussions and is not correct here.]

[44] to induce double-strand break (DSB) -> to induce a double-strand break (DSB)

[139] to be the top 2,000 -> to be in the top 2,000

[204] a piece of the genome that harbor -> a piece of the genome that harbors

[205ff & 229ff] [Remove redundancy and verbatim text copies for the two sections:]

[205ff] "provides a general landscape 206 that, when a particular PAM is widely applied to high-throughput genome editing, more 207 genetic effects could be altered if the PAM contains more G/C bases."

[229ff] "provide a general landscape for high-throughput genome editing that, when a particular PAM is widely applied across the human genome, more genetic effects could be altered or introduced if the PAM contains more G/C bases."

[234] even in the other species -> even in other species

[276] Same as the autosomes -> In line with procedures followed in the analysis of autosomes

Point-to-point responses to reviewers

for “Contribution of CRISPRable DNA to human complex traits” (final revision)

Aug 8, 2022

Contents

1 Responses to Reviewer #3

1

1 Responses to Reviewer #3

The current correlation analysis as stated in lines 138 to 140 describes the behaviour across all PAMs. In line with this and going through the supplementary “PAM content vs GC%” plots one more time, it strikes me that an “r” for a simple correlation analysis per PAM would likewise be highly informative for each PAM and more so than enumerating the “number of top-2000 PAM segments that are not in the top-2000 GC-rich segments.” By comparison, the present enumeration of non-corresponding top-2000 segments, while informative, feels somewhat construed. More importantly, eyeballing over the graphs, I would naively assume that a correlation of PAM content vs GC% (for the 18,763 segments) would e.g. results in an r approaching 1 for the “GG” PAM and would result in an r approaching -1 for the “AAA” PAM. The r could then be taken as an approximate single measure for a given PAM of how suitable it would be for the targeting of GC rich (and thus by inference “trait-relevant”) sections of the genome.

At this point it would be churlish to insist on this analysis, but if the authors agree that r as a measure of PAM content vs GC% per segment would make interpretation of their data more intuitive (and thus more citable), I invite them to add this number to each alignment to strengthen the article.

A: We agree that the correlation between GC and PAM contents in each segment is informative to measure how suitable a given PAM is to target GC-rich regions, so we added Pearson’s correlation coefficients in Supplementary Figures 2-77 following your kind suggestion. We include these supplementary figures not only to show the correlations between GC and PAM contents in each segment but also to show how the

PAMs are distributed across the genome. We also wanted to point out that there are exceptions where a PAM with high GC content may exist more frequently in a relatively low GC content segment (**Fig. 2**, lines 140-144).

[Title] Contribution of CRISPRable DNA on human complex traits -> Contribution of CRISPRable DNA to human complex traits [The preposition “on” is really only used for content contributions e.g. for discussions and is not correct here.]

[44] to induce double-strand break (DSB) -> to induce a double-strand break (DSB)

[139] to be the top 2,000 -> to be in the top 2,000

[204] a piece of the genome that harbor -> a piece of the genome that harbors

[205ff & 229ff] [Remove redundancy and verbatim text copies for the two sections:] [205ff] “provides a general landscape 206 that, when a particular PAM is widely applied to high-throughput genome editing, more 207 genetic effects could be altered if the PAM contains more G/C bases.” [229ff] “provide a general landscape for high-throughput genome editing that, when a particular PAM is widely applied across the human genome, more genetic effects could be altered or introduced if the PAM contains more G/C bases.”

[234] even in the other species -> even in other species

[276] Same as the autosomes -> In line with procedures followed in the analysis of autosomes the present study.”

A: All revised. Thanks again for providing such a detailed list of comments!

Prof. Xia Shen, on behalf of the co-authors